# Unsupervised classification of simulated magnetospheric regions

Maria Elena Innocenti[1], Jorge Amaya[2], Joachim Raeder[3], Romain Dupuis[2], Banafsheh Ferdousi[3], and Giovanni Lapenta[2]

[1]Institut für Theoretische Physik, Ruhr-Universität Bochum, Bochum, Germany
[2]Centre for mathematical Plasma Astrophysics, Department of Mathematics, KU Leuven, Leuven, Belgium
[3]Institute for the Study of Earth, Oceans and Space, University of New Hampshire, Durham, NH, USA

**Correspondence:** Maria Elena Innocenti (mariaelena.innocenti@rub.de)

**Abstract.**

In magnetospheric missions, burst mode data sampling should be triggered in the presence of processes of scientific or operational interest. We present an unsupervised classification method for magnetospheric regions, that could constitute the first-step of a multi-step method for the automatic identification of magnetospheric processes of interest. Our method is based on Self Organizing Maps (SOMs), and we test it preliminarily on data points from global magnetospheric simulations obtained with the OpenGGCM-CTIM-RCM code. The dimensionality of the data is reduced with Principal Component Analysis before classification. The classification relies exclusively on *local* plasma properties at the selected data points, without information on their neighborhood or on their temporal evolution. We classify the SOM nodes into an automatically selected number of classes, and we obtain clusters that map to well defined magnetospheric regions. We validate our classification results by plotting the classified data in the simulated space and by comparing with K-means classification. For the sake of result interpretability, we examine the SOM feature maps (magnetospheric variables are called *features* in the context of classification), and we use them to unlock information on the clusters. We repeat the classification experiments using different sets of features, we quantitatively compare different classification results, and we obtain insights on which magnetospheric variables make more effective features for unsupervised classification.

## 1  Introduction

The growing amount of data produced by measurements and simulations of different aspects of the heliospheric environment has made it fertile ground for applications rooted in Artificial Intelligence, AI, and Machine Learning, ML  (Bishop, 2006; Goodfellow et al., 2016). The use of ML in space weather nowcasting and forecasting is addressed in particular in Campo-reale (2019). ML methods promise to help sorting through the data, find unexpected connections, and can hopefully assist in advancing scientific knowledge.

Much of the AI/ML effort in space physics is directed at the Sun itself, either in the form of classification of solar images (Armstrong and Fletcher, 2019; Love et al., 2020) or for the forecast of transient solar events (see Bobra and Couvidat (2015); Nishizuka et al. (2017); Florios et al. (2018) and references therein). This is not surprising, since the Sun is the driver

of the heliospheric system and the ultimate cause of space weather (Bothmer and Daglis, 2007). Solar imaging is also one of the fields in science where data is being produced at an increasingly faster rate, see Figure 1 in Lapenta et al. (2020).

Closer to Earth, the magnetosphere has been sampled for decades by missions delivering an ever-growing amount of data, although magnetospheric missions are still far away from producing as much data as solar imaging. The four-spacecraft Cluster mission (Escoubet et al., 2001) has been investigating the Earth's magnetic environment and its interaction with the solar wind for over 20 years. Laakso et al. (2010), introducing a publicly accessible archive for high-resolution Cluster data, expected it to exceed 50 TB. The Magnetospheric Multiscale Mission (MMS) (Burch et al., 2016) is a four-spacecraft mission launched in 2015 with the objective of investigating the micro-physics of magnetic reconnection in the terrestrial magnetotail and magnetopause. It collects a combined volume of ∼100 gigabits per day of particle and field data, of which only about 4 can be transmitted to the ground due to downlink limitations (Baker et al., 2016). The Time History of Events and Macroscale Interactions during Substorms (THEMIS) mission (Angelopoulos, 2009) is composed of 3+2 spacecrafts launched in 2007 to investigate the role of magnetic reconnection in triggering substorm onset. It produces ∼2.3 gigabits data per day. Comparing THEMIS and MMS, we see an increase in the volume of data produced of almost two orders of magnitude in just 8 years.

Several studies have applied classification techniques to different aspects of the near-Earth space environment. Supervised classification has been extensively used for the detection and classification of specific processes or regions. An incomplete list of recent examples includes the detection and classification of magnetospheric Ultra Low Frequency Waves (Balasis et al., 2019), the detection of quasi-parallel and quasi-perpendicular magnetosheath jets (Raptis et al., 2020), and the detection of magnetopause crossings (Argall et al., 2020). Supervised techniques have also been used for the classification of "large scale" geospace regions, such as the solar wind, the magnetosheath, the magnetosphere and ion foreshock in Olshevsky et al. (2019), the solar wind, ion foreshock, bow shock, magnetosheath, magnetopause, boundary layer, magnetosphere, plasma sheet, plasma sheet boundary layer, lobe in Breuillard et al. (2020), magnetosphere, magnetosheath and solar wind in MMS data in Nguyen et al. (2019); da Silva et al. (2020). In the context of kinetic physics, Bakrania et al. (2020) have applied dimensionality reduction and unsupervised clustering methods to the magnetotail electron distribution in pitch angle and energy space, and have identified eight distinct groups of distributions, related to different plasma dynamics.

Almost all the magnetospheric classification studies mentioned above use supervised classification methods. Supervised classification relies on the previous knowledge and input of human experts. The accuracy of these methods comes from the use of known correlations between inputs and the corresponding output classes, presented to the algorithm during the training phase. Supervised classification is sometimes made unpractical by the need to label large amounts of data for the training phase. In magnetospheric studies this is often less of an issue due to the widespread availability of labelled magnetospheric data sets such as, for example, data tagged by the MMS "Scientist In The Loop" (Argall et al., 2020). However, the personal biases of the particular scientist labelling the data limits the ability of the algorithms to detect new and previously unknown pattern in the data.

On the other hand, unsupervised machine learning models can discern patterns in large amounts of *unlabeled* data without human intervention, and the associated biases. Clustering techniques separate data points into groups that share similar properties. Each cluster is represented by a mean value or by a centroid point. A good clustering technique will produce a

set of centroids with a distribution in data space that resembles closely the data distribution of the full data set. This allows

to differentiate groups of points in the data space and to identify dense distributions. Unsupervised methods are particularly useful to discover data patterns in very large data sets composed of multidimensional (i.e., characterized by multiple variables) properties, without having any kind of input from human experts. This is an unbiased, automatic, approach to discover hidden information in large simulation and observation data sets. In other words, while supervised ML can be useful in applying known methods to a broader set of data, unsupervised ML holds the promise of achieving true discovery or new insight.

Here, we explore an unsupervised classification method for simulated magnetospheric data points based on Self Organizing Maps (SOMs). Simulated data points are used to train a SOM, whose nodes are then clustered into an optimal number of classes. A posteriori, we try to map these classes to recognizable magnetospheric regions.

The objective of this work is to understand if the method we propose can be a viable option for the classification of magnetospheric spacecraft data into large scale magnetospheric regions. We also aim at gaining insights into the specifics of the

magnetospheric system (which are the best magnetospheric variables to use to train the classifiers? Which is the optimal cluster number?), that can later help us extending our work to spacecraft data. At the current stage, we move our first steps in the controlled and somehow easily understandable environment of simulations, where time-space ambiguities are eliminated, and one can validate classification performance by plotting the classified data point in the simulated space. In Section 2, we recall the main characteristics of the OpenGGCM-CTIM-RCM code, the global MagnetoHydroDynamics (MHD) code used in this

study to simulate the magnetosphere, and we show some preliminary analysis of the data we obtain. A brief description of the SOM algorithm is provided in Section 3. In Section 4 we illustrate our classification methodology. In Section 4.1 we analyze a classification experiment done with one particular set of magnetospheric features (we call magnetospheric variable *features* in the context of classification), and we realize that our unsupervised classification results agree to a surprisingly degree with with a human would do. In Section 4.2, we focus on model validation, and in particular on temporal robustness and on comparison

with another unsupervised classification method. In Section 4.3, we examine different sets of feature for the SOM training. We obtain alternative (but still physically significant) classification results, and some insights into what constitutes a "good" set of features for our classification purposes. Discussions and conclusions follow.

Further information of interest is provided in Appendix A, where we report on manual exploration of the SOM hyper-parameter space, and in Appendix B, where we assess how robust our classification method is by changing the number of

K-means cluster used to classify the SOM nodes.

## 2   Global magnetospheric simulations

The global magnetospheric simulations are produced with the OpenGGCM-CTIM-RCM code, a MHD-based model that simulates the interaction of the solar wind with the magnetosphere-ionosphere-thermosphere system. OpenGGCM-CTIM-RCM is

available at the Community Coordinated Modeling Center at NASA/GSFC for model runs on demand. A detailed description of the model and typical examples of OpenGGCM applications can be found in Raeder (2003), Raeder et al. (2001b), Raeder

and Lu (2005), Connor et al. (2016), Raeder et al. (2001a), Ge et al. (2011), Raeder et al. (2010), Ferdousi and Raeder (2016), Dorelli (2004), Raeder (2006), Berchem et al. (1995), Moretto et al. (2006), Vennerstrom et al. (2005), Anderson et al. (2017), Zhu et al. (2009), Zhou et al. (2012), Shi et al. (2014), to name a few. Of particular relevance to this study, OpenGGCM-CTIM-

RCM simulations have recently been used for a Domain of Influence Analysis, a technique rooted in Data Assimilation that can be used to understand what are the most promising locations for monitoring (i.e. spacecraft placing) in a complex system such as the magnetosphere (Millas et al., 2020).

OpenGGCM-CTIM-RCM uses a stretched Cartesian grid (Raeder, 2003), which in this work has 325x150x150 cells, sufficient for our large scale classification purposes, while running for few hours on a modest number of cores. The point density

increases in the Sunwards direction and in correspondence with the magnetospheric plasma sheet, the "interesting" region of the simulation for our current purposes. The simulation extends from $-3000\ R_E$ to $18\ R_E$ in the Earth-Sun direction, from $-36\ R_E$ to $+36\ R_E$ in the y and z direction. $R_E$ is the Earth's mean radius, the Geocentric Solar Equatorial (GSE) coordinate system is used in this study.

In this work, we do not classify points from the entire simulated domain. We focus on a subset of the points with coordinates

$-41 < x/R_E < 18$, i.e. the magnetosphere / solar wind interaction region and the near magnetotail.

The OpenGGCM-CTIM-RCM boundary conditions require the specification of the three components of the solar wind velocity and magnetic field, the plasma pressure and the plasma number density at 1 AU. Boundary conditions in the sunward direction vary with time. They are interpolated to the appropriate simulated time from ACE observations (Stone et al., 1998), and applied identically to the entire Sunward boundary. At the other boundaries, open boundary conditions (i.e., zero normal

derivatives) are applied, with appropriate corrections to satisfy the $\nabla \cdot \mathbf{B} = 0$ condition.

For this study we initialize our simulation with solar wind conditions observed starting from May $8^{th}$, 2004, 09:00 UTC, denoted as $t_0$. After a transient, the magnetosphere is formed by the interaction between the solar wind and the terrestrial magnetic field.

We classify simulated data points from the time $t_0 + 210$ minutes, when the magnetosphere is fully formed. We later compare

our results with earlier and later times, $t_0 + 150$ and 225 minutes. In Figure 1, we show significant magnetospheric variables, at $t_0 + 210$ minutes, in the $xz$ plane at $y/R_E = 0$ (meridional plane): the three components of the magnetic field $B_x$, $B_y$, $B_z$, the three velocity components $v_x$, $v_y$, $v_z$, the plasma density $n$, pressure $pp$, temperature $T$, the Alfvén speed $v_A$, the Alfvénic Mach number $M_A$, the plasma $\beta$. Magnetic field lines (more precisely, lines tangential to the field direction within the plane) are drawn in black. Panels g to l are in logarithmic scale.

We expect the algorithm to identify well-known domains such as pristine solar wind, magnetosheath, lobes, inner magnetosphere, plasma sheet, and boundary layers, which can be clearly identified in these plots. The classification is done for points from the entire 3D volume, and not only for 2D cuts such as the one shown.

In Figure 2, we depict the violin plots for the variables in Figure 1. Violin plots are useful tools to visualize at a glance the distribution of feature values, as they depict the probability density of the data at different values. In violin plots, the shape

of the violin depicts the frequency of occurrence of each feature: the "thicker" regions of the violin are where most of the observations lie. The white dot, the thick black vertical lines and the thin vertical lines (the "whiskers") separating the blue

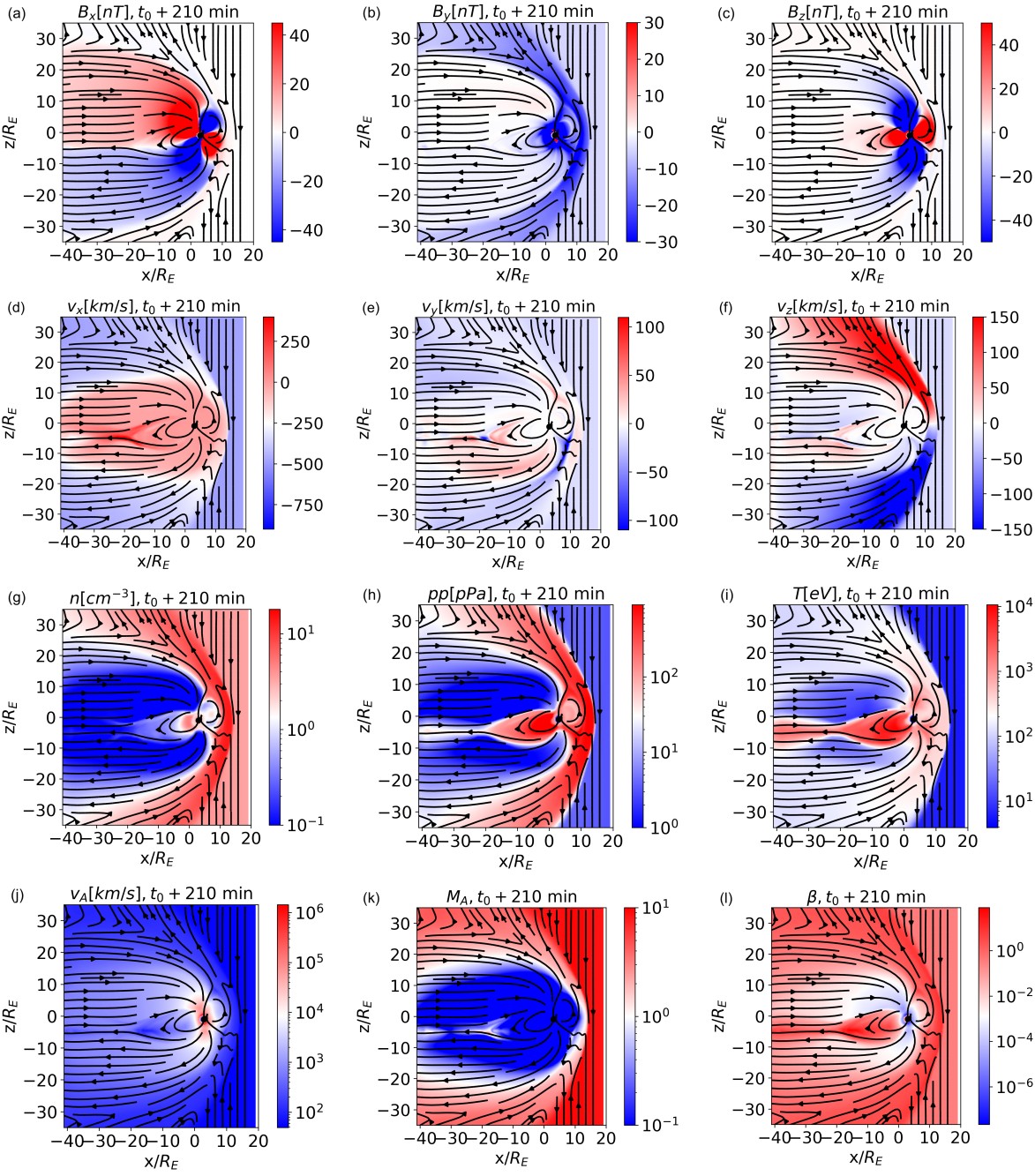

**Figure 1.** Simulated magnetospheric variables, in the $y/R_E = 0$, meridional, plane and at $t_0$+ 210 minutes. The different magnetospheric regions are evident. Magnetic field lines are depicted as black arrows.

and orange distributions depict the median, the interquartile range, and the 95 % confidence interval respectively. 50 % of the data lie in the region highlighted by the thick black vertical lines. 95 % of the data lie in correspondence of the whiskers. The left and right sides of the violins depict distributions at different times. The left side, in blue, depicts data points from $t_0 + 210$ minutes. The right side, in orange, depicts a data set composed of points from multiple snapshots, $t_0 + 125, 175, 200$ minutes, and intends to give a visual assessment of the variability of the distribution of magnetospheric properties with time. The width of the violins are normalized to the number of points in each bin.

In the simulations, points closer to Earth correctly exhibit very high magnetic field values, up to several $\mu$T. In the violin plots, for the sake of visualization, the magnetic field components of points with $|B| > 100 \, nT$ have been clipped to $\sqrt{100^2}/3 \, nT$, multiplied by their respective sign (hence the accumulation of points at $\pm\sqrt{100^2}/3 \, nT$ in the magnetic field components). The multi-peaked distribution of several of the violins reflect the variability of these parameters across different magnetospheric environments. Multi-peaked distributions bode well for classification, since they show that the underlying data can be inherently divided in different classes.

## 3 Self Organizing Maps: a recap

To classify magnetospheric regions, we use Self Organizing Maps (SOMs), an unsupervised ML technique. Self Organizing Maps (Kohonen, 1982; Villmann and Claussen, 2006; Kohonen, 2014; Amaya et al., 2020), also known as Kohonen maps or self organizing feature maps, are a clustering technique based on a neural network architecture. SOMs aim at producing an *ordered* representation of data, which in most cases has lower dimensionality with respect to the data itself. "Ordered" is a key word in SOMs. The topographical relations between the trained SOM nodes are expected to be similar to those of the source data: data points that map to "nearby" SOM nodes are expected to have "similar" features. Each SOM node then represents a local average of the input data distribution, and nodes are topographically ordered according to their similarity relations (Kohonen, 2014).

A SOM is composed of :

- a (usually) two dimensional lattice of $Lr \times Lc = q$ nodes, with $Lr$ and $Lc$ the number of rows and columns. This lattice, also called a map, is a structured arrangement where all nodes are located in fixed positions, $\mathbf{p}_i \in \mathbb{R}^2$, and associated with a single *code word*, $\mathbf{w}_i$. As it is often done with two-dimensional SOM lattices, the nodes are organized in an hexagonal grid (Kohonen, 2014).

- a list of q *code words* $\mathbf{W} = \{\mathbf{w}_i \in \mathbb{R}^n\}_{i=0..q-1}$, where $n$ is the number of features associated to each data point (and hence to each code word). $n$ is therefore the number of plasma variables that we select, among the available ones, for our classification experiment. Each $\mathbf{w}_i$ is associated with a map node $\mathbf{p}_i$

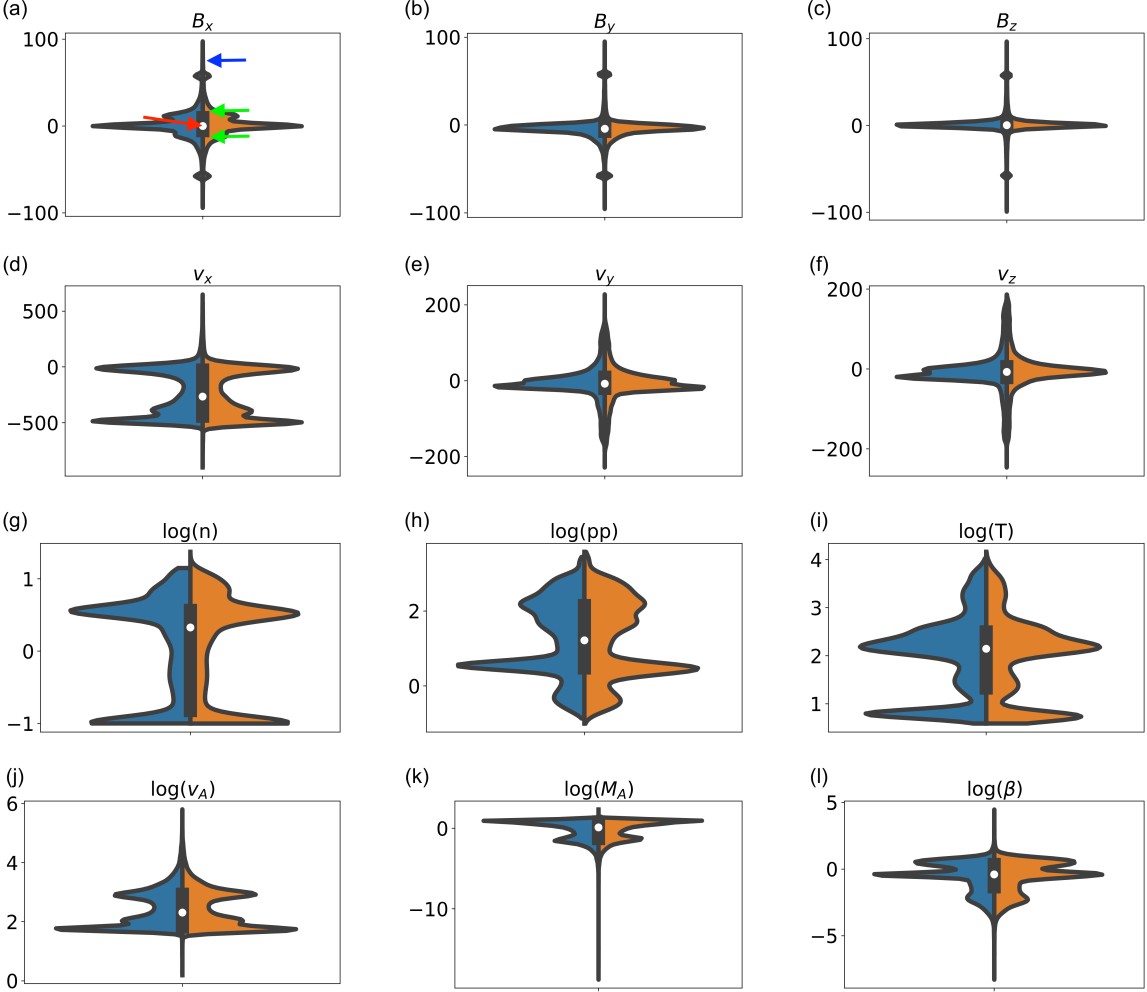

**Figure 2.** Violin plots of the data sets extracted from the magnetospheric simulations. The left sides of the violins, in blue, are data points at $t_0 + 210$ minutes, the right, in orange, are from $t_0 + 125, 175, 200$ minutes. In the $B_x$ plot, the red, green, blue arrows point at the median, first and third quartiles, whiskers, respectively.

Each of the $m$ input data points is a data entry $\mathbf{x}_\tau \in \mathbb{R}^n$. Notice that, in the rest of the manuscript, we will use terms such as "data point", "data entry", "input point" interchangeably. Given a data entry $\mathbf{x}_\tau$, the closest code word in the map, $\mathbf{w}_s$ (Eq. 1), is called the *winning element*, and the corresponding map node $\mathbf{p}_s$ is the Best Matching Unit (BMU):

$$\mathbf{w}_s = \underset{\mathbf{w}_i \in \mathbf{W}}{\arg\min} \left( \|\mathbf{x}_\tau - \mathbf{w}_i\| \right) \tag{1}$$

$||.||$ is the distance metric. In this work, we use the Euclidean norm.

SOMs learn by moving the winning element *and neighboring nodes* closer to the data entry, based on their relative distance, and on a iteration number-dependent learning rate $\epsilon(\tau)$, with $\tau$ the progression of samples being presented to the map for training: the feature values of the winning element are altered as to reduce the distance between the "updated" winning element and the data entry. The peculiarity of the SOMs is that a single entry is used to update the position of several code words in feature space, the winning nodes and its nearest neighbors: code words move towards the input point at a speed proportional to the distance of their correspondent lattice node position to the *winning node*.

It is useful to compare the learning procedure in SOMs and in another, perhaps more known, unsupervised classification method, K-means (Lloyd, 1982). Both SOMs and K-means classification identify and modify the best matching unit for each new input. In K-means, only the winner node is updated. In SOMs, the winner node *and its neighbors* are updated. This is done to obtain an ordered distribution: nearby nodes, notwithstanding their initial weights, are modified during training as to become more and more similar.

At every iteration of the method, the code words of the SOM are shifted according to the following rule:

$$\Delta\mathbf{w}_i = \epsilon(\tau)h_\sigma(\tau_i, i, s)(\mathbf{x}_\tau - \mathbf{w}_i) \tag{2}$$

with $h_\sigma(\tau, i, j)$ defined as the lattice neighbor function:

$$h_\sigma(\tau, i, j) = e^{-\frac{\|\mathbf{P}_i - \mathbf{P}_j\|^2}{2\sigma(\tau)^2}}, \tag{3}$$

where $\sigma(\tau)$ is the iteration-number dependent lattice neighbor width. The training of the SOM is an iterative process. At each iteration a single data entry is presented to the SOM and code words are updated accordingly. The radius of the neighboring function $\sigma(\tau)$ determines how far from the winning node the update introduced by the new input will extend. The learning rate $\epsilon(\tau)$ gives a measure of the magnitude of the correction. Both are slowly decreased with the iteration number. At the beginning of the training, the update introduced by a new data input will extend to a large number of nodes (large $\sigma$), which are significantly modified (large $\epsilon$), since it is assumed that the map node do not represent well the input data distribution. At large iteration number, the nodes are assumed to have already become more similar to the input data distribution, and lower $\sigma$ and $\epsilon$ are used for "fine tuning".

In this work, we choose to decrease $\sigma$ and $\epsilon$ with the iteration number. Another option, which we do not explore, is to divide the training into two stages, coarse ordering and final convergence, with different values of $\sigma$ and $\epsilon$.

However small, $\sigma$ has to be kept larger than 0, otherwise only the winning node is updated, and the SOM loses its ordering properties (Kohonen, 2014).

This learning procedure ensures that neighboring nodes in the lattice are mapped to neighboring nodes in the $n$-dimensional feature space. The 2D maps obtained can then be graphically displayed, allowing to visually recognize patterns in the input features and to group together points that have similar properties, see Figure 5.

**Table 1.** Combination of features ("Cases") used for the different classification experiments. We mark in bold differences with respect to F1, our reference feature set.

| Case | Features |
|:---:|:---:|
| F1 | $B_x, B_y, B_z, v_x, v_y, v_z$, log(n), log(pp), log(T) |
| F1-NL | $B_x, B_y, B_z, v_x, v_y, v_z$, **n, pp, T** |
| F2 | $B_x, B_y, B_z, v_x, v_y, v_z$, **log(n)**, log(pp), log(T) |
| F3 | $B_x, B_y, B_z$, **$\mathbf{v}_x$**, $v_y, v_z$, log(n), log(pp), log(T) |
| F4 | $B_x, B_y, B_z, v_x, v_y, v_z$, log(n), log(pp), log(T), **log($\mathbf{v}_A$)** |
| F5 | $B_x, B_y, B_z, v_x, v_y, v_z$, log(n), log(pp), log(T), **log($\mathbf{M}_A$)** |
| F6 | $B_x, B_y, B_z, v_x, v_y, v_z$, log(n), log(pp), log(T), **log($\beta$)** |
| F7 | **$\mathbf{B}_x, \mathbf{B}_y, \mathbf{B}_z$**, $v_x, v_y, v_z$, log(n), log(pp), log(T) |
| F8 | **$\mathbf{B}_x, \mathbf{B}_y, \mathbf{B}_z$ (clipped)**, $v_x, v_y, v_z$, log(n), log(pp), log(T) |
| F9 | **\|B\| (clipped)**, $v_x, v_y, v_z$, log(n), log(pp), log(T) |
| F10 | **log\|B\|**, $v_x, v_y, v_z$, log(n), log(pp), log(T) |

The main metric for the evaluation of the SOM is the *quantization error*, which measures the average distance between each of the $m$ entry data points and its BMU, and hence how closely the map reflects the training data distribution.

$$Q_E = \frac{1}{m} \sum_{i=1}^{m} \|\mathbf{x}_i - \mathbf{w}_{s|\mathbf{x}_i}\| \tag{4}$$

## 4 Methodology and results

For our unsupervised classification experiments, we initially focus on a single temporal snapshot of the OpenGGCM-CTIM-RCM simulation, $t_0 + 210$ minutes. Although the simulation domain is much larger, we restrict our input data set to the points with $-41 < x/R_E < 18$ (see Figure 1), since we are particularly interested in the magnetospheric regions more directly shaped by interaction with the solar wind. We select 1 % of the 5557500 data points at $x/R_E > -41$, $t = t_0 + 210$ minutes as the training data set. The selection of these points is randomized, and the seed of the random number generator is fixed to ensure that results can be reproduced. Tests with different seeds and with an higher number of training points did not give significantly different classification results.

In Figure 3, panel a, we plot the correlation matrix of the training data set. This and all subsequent analyses, unless otherwise specified, are done with the feature list labeled as F1 in Table 1: the three components of the magnetic field and of the velocity, the logarithm of the density, pressure, temperature. In Table 1, we describe the different sets of features used in the classification experiments described in Section 4.3. Each set of feature is assigned an identifier ("Case"); the list of magnetospheric variables used in each case are listed under "Features". Differences with respect to F1 are marked in red.

The correlation matrix shows the correlation coefficients between a variable and all others, including itself (the correlation coefficient of a variable with itself is of course 1). We notice in the bottom right of the matrix that correlation is high between logarithm of density, logarithm of pressure, logarithm of temperature and velocity in the Earth-Sun direction. This suggests that a lower-dimension feature set can be obtained, that still expresses a high percentage of the original variance. Using a lower dimensional training data set is desirable, since it reduces the training time of the map.

At this stage of our investigation, we use Principal Component Analysis (PCA) (Shlens, 2014) as a dimensionality reducing tool. More advanced techniques, and in particular techniques that do not rely on linear correlation between the features, are left as future work.

First, the variables are scaled between two fixed numbers, here 0 and 1, to prevent those with larger ranges from dominating the classification. Then, we use PCA to extract linearly independent Principal Components, PCs, from the set of original variables. We keep the first three PCs, which express 52 %, 35 % and 5.4 % of the total variance, retaining 93% of the initial variance. We plot in Figure 3, panel b to d, left blue half-violins, the violin plots of these scaled components. For a visual assessment of temporal variability in the simulations we show in the right orange half-violins the first three PCs of the mixed-time data set, where data points are taken at $t_0 + 125, 175, 200$ minutes. We see difference, albeit small, between the two sets, which explain the different classification results with fixed and mixed time data sets that we discuss in Section 4.3. Notice, by comparing the blue and orange half-violins in panel b, that PC0 is "rotated" around the median value in the two data sets, which is possible for components reconstructed through linear PCA.

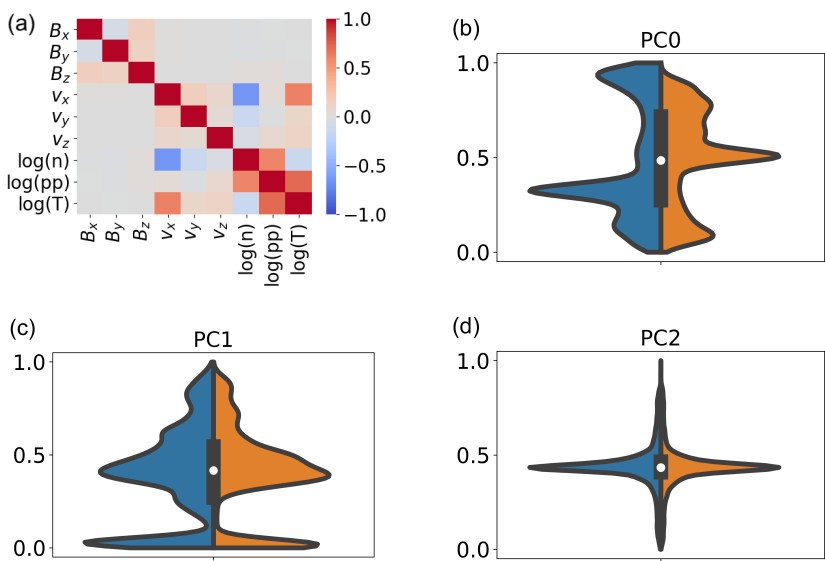

**Figure 3.** Correlation plot for the fixed-time training data set at time $t_0 + 210$ minutes (panel a), violin plots of the first three PCs after PCA for the fixed- (left, blue half-violins) and mixed- (right, orange half-violins) time data sets (panel b to d).

**Table 2.** Eigenvectors associated with the first three PCs (rows), for each of the F1, clipped F1, clipped F9 features in Table 1 (columns). The first most relevant features for each PC are marked in bold fonts.

| | $|B|$ | $B_x$ | $B_y$ | $B_z$ | $v_x$ | $v_y$ | $v_z$ | $log(n)$ | $log(pp)$ | $log(T)$ |
|---|---|---|---|---|---|---|---|---|---|---|
| | | | | F1 feature set | | | | | | |
| PC0 | - | 1.73e-04 | 2.83e-04 | -6.50e-04 | **2.47e-01** | 4.57e-02 | 5.89e-03 | **-8.55e-01** | **-4.52e-01** | -3.20e-02 |
| PC1 | - | -7.67e-06 | -7.51e-05 | 1.59e-04 | **3.67e-01** | 5.77e-02 | 5.30e-02 | -1.87e-01 | **5.07e-01** | **7.53e-01** |
| PC2 | - | -2.12e-04 | 9.58e-06 | -3.81e-04 | -9.59e-03 | **-9.80e-01** | **-1.75e-01** | -6.78e-02 | **1.71e-02** | 6.38e-02 |
| | | | | F8 feature set - B clipped | | | | | | |
| PC0 | - | 4.44e-02 | 6.16e-02 | -4.59e-02 | **2.43e-01** | 4.55e-02 | 5.64e-03 | **-8.51e-01** | **-4.54e-01** | -3.73e-02 |
| PC1 | - | -1.78e-02 | -1.42e-02 | 4.62e-02 | **3.7e-01** | 5.8e-02 | 5.29e-02 | -1.96e-01 | **5.01e-01** | **7.51e-01** |
| PC2 | - | **-1.74e-01** | -2.61e-02 | -3.54e-02 | -2.1e-02 | **-9.59e-01** | **-1.94e-01** | -7.78e-02 | 1.39e-02 | 6.60e-02 |
| | | | | F9 feature set - B clipped | | | | | | |
| PC0 | 1.12e-01 | - | - | - | **2.8e-01** | 4.94e-02 | 1.00e-02 | **-8.61e-01** | **-4.05e-01** | 2.98e-02 |
| PC1 | **4.72e-01** | - | - | - | 3.11e-01 | 4.09e-02 | 4.43e-02 | -4.64e-02 | **4.98e-01** | **6.53e-01** |
| PC2 | **8.57e-01** | - | - | - | -2.67e-02 | -6.25e-02 | -3.68e-02 | **1.93e-01** | -2.31e-01 | **-4.11e-01** |

To investigate which of the features contribute most to each PC, we print in Table 2, section "F1 feature set", the eigenvectors associated with the first three PCs (rows). Each column correspond to one feature. The three most relevant features for each PC are marked in bold fonts.

The three most significant F1 features for PC0 are the logarithm of the density, of the pressure and the velocity in the $x$ direction; for PC1 the logarithm of the temperature, of the pressure and the velocity in the $x$ direction; for PC2 the velocity in the $y$ direction, in the z direction and the logarithm of the plasma pressure. We see that the three magnetic field components rank the lowest in importance for all the three PCs.

This last result is at a first glance quite surprising, given the fundamental role of the magnetic field in magnetospheric dynamics. We can explain it looking at the violin plots in Figure 2. There, we see that the magnetic field distributions are quite "simple" when compared to the multi-peak distributions of more significant features such as density, pressure, temperature, $v_x$. Still, one may argue that the very high values of the magnetic field close to Earth distort the magnetic field component distributions, and reduce their weight in determining the PCs. In Table 2, section "F8 feature set - B clipped", we repeat the analysis clipping the magnetic field values as in Figure 2. The intention of the clipping procedure is to cap the maximum magnitude of the magnetic field module to 100 nT, while retaining information on the sign of each magnetic field component. Also now, the magnetic field does not contribute significantly in determining the PCs.

In Table 2, section "F9 feature set - B clipped", we list the eigenvectors associated with the first three PCs for the F9 feature set. We see that now the clipped magnetic field magnitude ranks higher than with "F1 feature set" and "F8 feature set - B clipped" in determining the PCs, and becomes relevant especially for PC1 and PC2. In the violin plots of the PCs for F9, not shown

here, we see that PC0 is not significantly different in F1 and F9, while PC1 and PC2 are. In particular, PC2 for F9 exhibits
more peaks than PC2 for F1.

The first three PCs obtained from the F1 feature list (without magnetic field clipping) are used to train a SOM. Each of the
data points is processed and classified separately, based solely on its local properties, at $t_0 + 210$ minutes. We consider this
local approach one of the strengths of our analysis method, which makes it particularly appealing for spacecraft on-board data
analysis purposes.

The procedure for the selection of the SOM hyper-parameters is described in Appendix A. At the end of it, we choose the
following hyper-parameters: $q = 10 \times 12$ nodes, initial learning rate $\epsilon_0 = 0.25$ and initial lattice neighbor width $\sigma_0 = 1$.

After the SOM is generated, its nodes are further classified using K-means clustering in a predetermined number of classes.
Data points are then assigned to the same cluster as their Best Matching Unit, BMU.

The overall classification procedure can then be summarized as follows:

1. data pre-processing: feature scaling, dimensionality reduction via PCA, scaling of the reduced values;

2. SOM training;

3. K-means clustering of the SOM nodes;

4. classification of the data points, based on the classification of their BMU.

It is useful to remark that, even if the same data are used to train different SOMs, the trained networks will differ due e.g. to
the stochastic nature of artificial neural networks and to their sensitivity to initial conditions. If the initial positions of the map
nodes are randomly set (as in our case), maps will evolve differently, even if the same data are used for the training.

To verify that our results do not correspond to local minima, we have trained different maps seeding the initial random node
distribution with different seed values. We have verified that the trained SOMs so generated give comparable classification
results, even if the nodes that map to the same magnetospheric points are located at different coordinates in the map. The
reason for this comparable classification results is that the 'net' created by a well-converged SOM will always have a similar
coverage and neighbouring nodes will always be located at similar distances with respect to their neighbours (if the training
data do not change). Hence, while the final map might look different, the classes and their properties will produce very similar
end results. We refer the reader to Amaya et al. (2020) for exploration of the sensitivity of the SOM method to the parameters
and to initial condition, and for a study of the rate and speed of convergence of the SOM.

Our maps are initialized with random node distributions. It has been demonstrated that different initialization strategies,
such as using as initial node values a regular sampling of the hyperplane spanned by the two principal components of the data
distribution, significantly speed up learning (Kohonen, 2014).

### 4.1 Classification results and analysis

We describe in this Section the results of a classification experiment with the feature set F1 from Table 1. After training the
SOM, we proceed to node clustering. The optimal number of K-means classes $k$ can be chosen examining the variation with $k$

of the Within Cluster Sum of Squares (WCSS), i.e. the sum of the squared distances from each point to their cluster centroid. The WCSS decreases as $k$ increases; the optimal value can be obtained using the Kneedle ("knee" plus "needle") class number determination method (Satopaa et al., 2011), that identifies the knee where the perceived cost to alter a system parameter is no longer worth the expected performance benefit. Here, the Kneedle method, Figure 4, panel a, gives $k = 7$ as the optimal cluster number, i.e. a representative and compact description of feature variability.

The clustering classification results can be plotted in 2D space. Figure 4 shows, in panel d and e, points with $-1 < y/R_E < 1$ and with $-1 < z/R_E < 1$, that we identify, for simplicity, with the meridional and equatorial plane respectively. The projected field lines are depicted in black. $k = 7$, as per the results of the Kneedle method. The points are depicted in colors, with each color representing a class of classified SOM nodes. The dot density changes in different areas of the simulation because the grid used in the simulation is stretched, with increasing points per unit volume in the Sunward direction and in the plasma sheet center. The (T) in the label is used to remind the reader that these points are the ones used for the training of the map. Plots of validation datasets will be labeled as (V).

The SOM map in panel c depicts the clustered SOM nodes. In panel b, the clusters are a posteriori mapped to different magnetospheric regions.

Comparing Figure 4 and Figure 1, we see that cluster 0, purple, corresponds to unshocked solar wind plasma. Cluster 4 (brown) and 1 (blue) map to shocked magnetosheath plasma just downstream the bow shock. Cluster 5 (orange) groups both points in the downwind supersonic magnetosheath, further downstream from the bow shock, and a few points *at* the bow shock. A possible explanation for this is that the bow shock is not in fact a vanishingly thin boundary, but has a finite thickness. The points within this region of space would present characteristics intermediate between the unshocked solar wind and the shocked plasma just downstream the bow shock, which are serendipitously very similar to those of other regions. Cluster 2, cyan, maps to boundary layer plasma. Cluster 3, green, corresponds to points in the inner magnetosphere.

The result of this unsupervised classification is actually quite remarkable, because it corresponds quite well to the "human" identification of magnetospheric regions developed over decades on the basis of analysis satellite data and understanding of physical processes. Here, instead, this very plausible classification of magnetospheric regions is obtained without human intervention.

In Figure 5 we plot the feature map associated with the classification in Figure 4. While a good correspondence between feature value at a SOM node and values at the associated data points can be expected for the features that contribute most to the first PCs, this cannot be expected with less relevant features, such as the three components of the magnetic field in this case (see Table 2, section "F1 feature set", and accompanying discussion). Keeping these considerations in mind while looking at the feature values across the map nodes in Figure 5, we see that they correspond quite well with what we expect from the terrestrial magnetosphere.

In particular, we see that the pristine solar wind (cluster 0 with $k = 7$ in Figure 4, in the bottom right corner of the map in Figure 5), is well separated in terms of properties from the neighboring regions, especially when considering $v_x$, plasma pressure and temperature. This is because the plasma upstream a shock is faster, lower-pressure and colder than the plasma downstream. In a shock, we expect higher density downstream the shock. We see that, of the three magnetosheath clusters

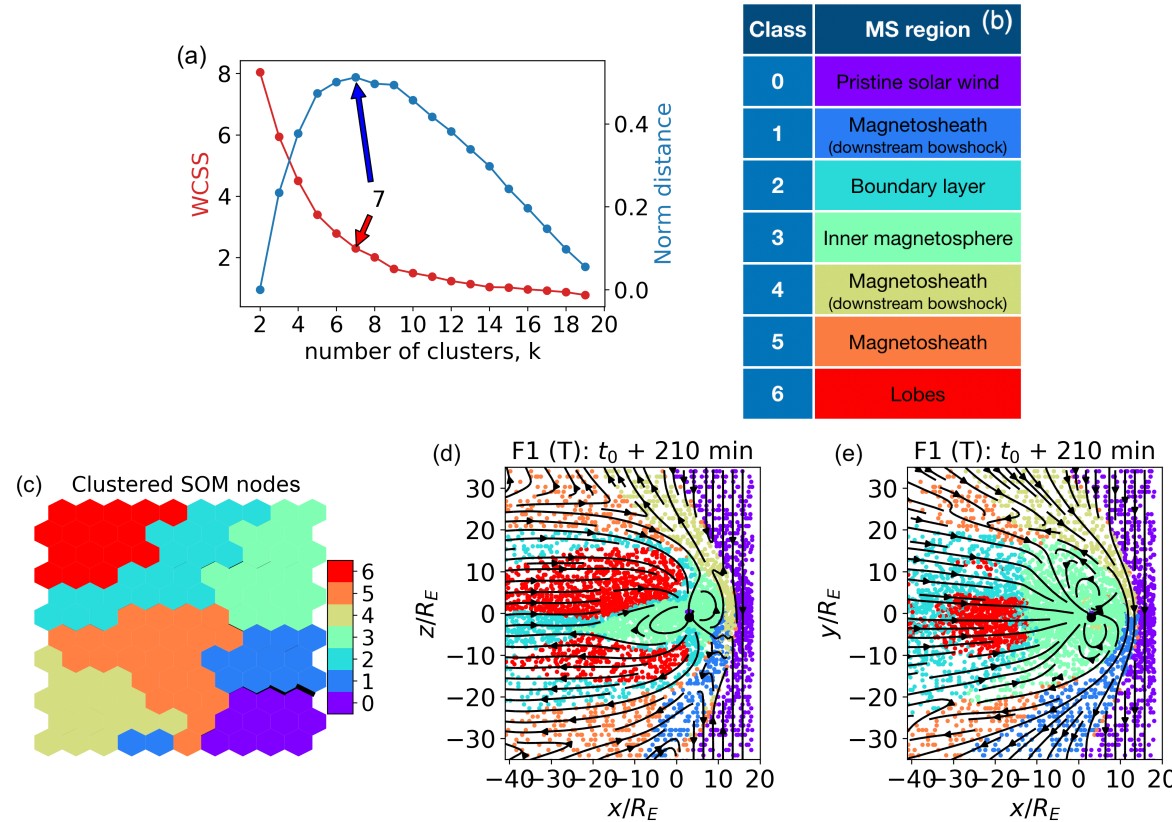

**Figure 4.** panel a: Kneedle determination of the optimal number of K-means clusters for the SOM nodes. WCSS (left axis) is the Within Cluster Sum of Squares, the maximum of the normalized distance (right axis) identifies the optimal cluster number, here $k = 7$. panel b: a posteriori class identification. panel c: clustered SOM nodes. Panel d and e: classified points in the meridional and equatorial planes respectively. In panel b to e, $k = 7$. The points depicted are the ones used for the training (T) of the map.

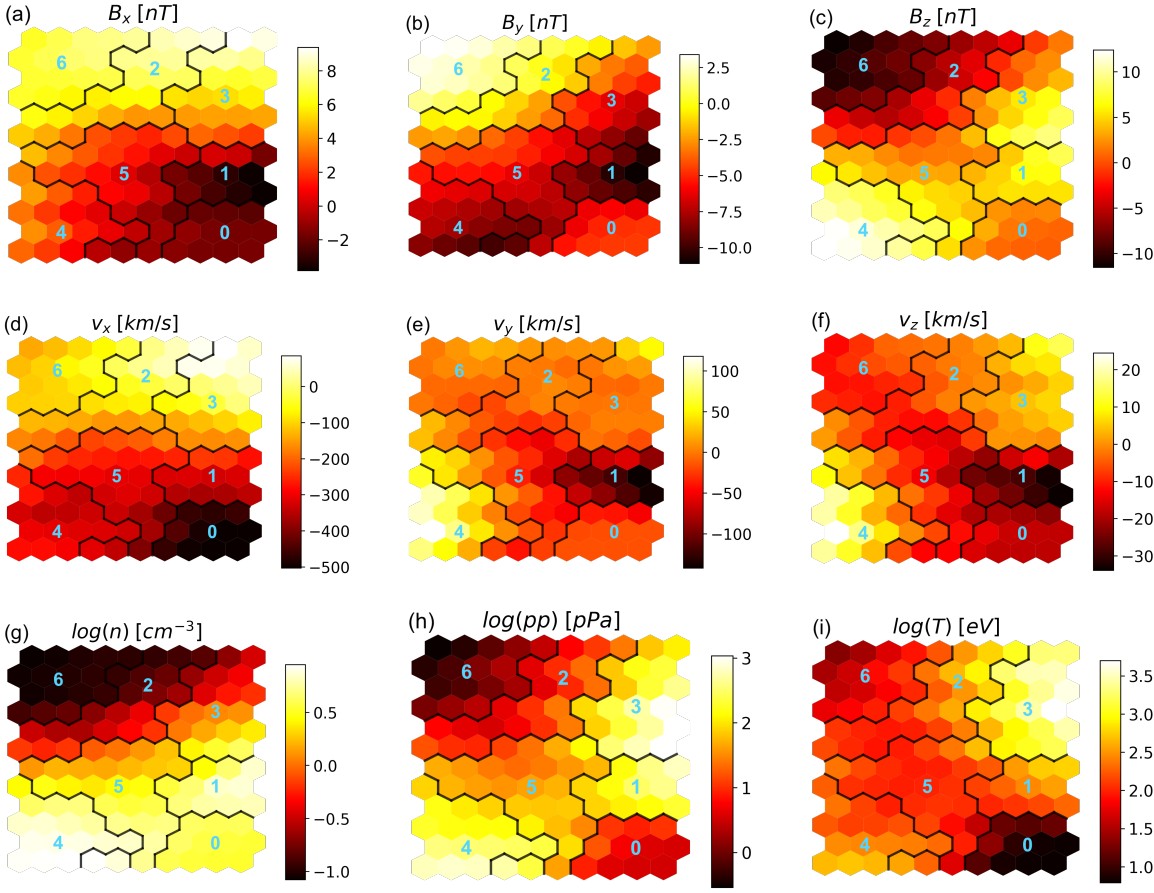

**Figure 5.** Distribution of the feature values in the SOM map, with $q = 10 \times 12$, $\sigma_0 = 1$, $\epsilon_0 = 0.25$. The cluster boundaries and numbers are for $k = 7$, Figure 4: cluster 0 corresponds to pristine solar wind, cluster 1, 4 and 5 to magnetosheath plasma, cluster 2 to boundary layers, cluster 3 to the inner magnetosphere, cluster 6 to the lobes.

(clusters 1, 4 and 5 in Figure 4; the cluster surrounding the bottom right cluster in Figure 5), the two mapping to regions just downstream the bow shock (cluster 1 and 4) have higher density than the solar wind cluster. When moving from cluster 1 and 4 towards the lobes, i.e. into cluster 5, the density decreases.

Cluster 1 and 4 are associated with magnetosheath plasma immediately downstream the bow shock. We see that their nodes have very similar values in terms of density, pressure, temperature, $v_x$, the quantities mainly associated to the downstream of the bow shock. They differ mainly in terms of the sign of the $v_z$ velocity components: the regions identified in Figure 4 as cluster 4 (1) have mainly positive (negative) $v_z$. This may be the reason why two nodes adjacent to cluster 4 but presenting $v_z < 0$ are carved out as cluster 1 in the map. We notice that, as a general rule, nodes belonging to the same cluster are expected to be contiguous in the SOM map, barring higher-dimension geometries which cannot be drawn in a 2D plane.

Other clusters that draw immediate attention are cluster 2 and 3, top right of the map, which are the only ones whose nodes include positive $v_x$ values (Sunward velocity). These clusters map to boundary layers and the inner magnetosphere: the $v_x > 0$ nodes are associated with the Earthwards fronts we see in Figure 1, panel d.

Finally, we remark on a seemingly strange fact: lobe plasma is clustered in cluster 6, which maps in Figure 5 to nodes associate to $B_x > 0$ only. We can explain this with the negligible role that $B_x$ has in determining the PCs for the F1 feature set (see discussion above in Section 4), and hence the map structure. We can expect that the feature map for features that rank higher in determining the PCs (here, density, pressure, temperature, $v_x$) will be more accurate than for lower-ranking features.

### 4.2 Model validation

In this Section, we address the robustness of the classification method when confronted with data from different simulated times, Section 4.2.1, and we compare against a different unsupervised classification method, pure K-means classification, in Section 4.2.2.

### 4.2.1 Robustness to temporal variations

In Figure 4 we plot classification results for the training set (T). Now, in Figure 6, we move to validation sets (V), composed of different points from the same simulated time as the training set (panel b and e, $t_0 + 210$ minutes), and of points from different simulated times (panel a and c, d and f $t_0 + 150$ and $t_0 + 225$ respectively). While panel b and e shows a straightforward sanity check, panel a and c aim at assessing how robust the classification method is to temporal variation. We want to verify how well classifiers trained at a certain time perform at different times, and in particular under different orientations of the geoeffective component of the Interplanetary Magnetic Field (IMF) $B_z$, which has well known and important consequences for the magnetic field structure. $B_z$ points Southwards at time $t_0 + 210$ and 225 minutes, Northwards at $t_0 + 150$ minutes.

The points classified in Figure 6, panel a to f, are pre-processed and classified not only with the same procedure, but also with the same scalers, SOM and classifiers described in Section 4, and trained on a subset of data from $t_0 + 210$ minutes. Panel a to c depicts points in the meridional plane, panel d to f in the equatorial plane.

While we can expect that the performance of classifiers trained at a single time will degrade when magnetospheric conditions

change, it is useful information to understand how robust to temporal variation they are, and which are the regions of the magnetosphere which are more challenging to classify correctly.

Examining Figure 6, panel b and e, we see that the classification results for the validation set at $t_0 + 210$ minutes excellently match those obtained with the training set, Figure 4. The classification outcomes at time $t_0 + 150$ minutes, panel a and d, are also well in line with time $t_0 + 210$. The biggest difference of plots at $t_0 + 150$ minutes with $t_0 + 210$ is in the Southern magnetosheath region just downstream the bow shock in the meridional plane: while this region is classified as cluster 1 at time $t_0 + 210$, it is classified at time $t_0 + 150$ mins as cluster 1 or 4, the other magnetosheath cluster downstream the bow shock,

mostly associated with the Northern magnetosheath at time $t_0 + 210$. In panel c, time $t_0 + 225$ minutes, all magnetosheath plasma downstream the bow shock is classified as cluster 1.

        This result can be easily explained. Cluster 1 and 4 both map to shocked plasma downstream the bow shock, i.e. regions with virtually identical properties in terms of the quantities that weigh the most in determining the PCs and therefore, arguably, the SOM structure: plasma density, pressure, temperature, $v_x$. The features that could help distinguishing between the North and

South sectors, $B_x$ and $v_z$, rank very low in determining the first PCs, and hence the SOM structure. On the other hand, exactly distinguishing via automatic classification between cluster 1 and 4 is not of particular importance, since the same physical processes are expected to occur in the two. Furthermore, a quick glance at the spacecraft spatial coordinates can clarify in which sector it is.

        Relatively more concerning is the fact that several points in the Sunwards inner magnetosphere at time $t_0 + 225$ minutes are

identified as inner magnetosheath plasma in Figure 6, panel c and f. While training the SOM with several feature combinations in Section 4.3, we notice that this particular region is perhaps the most difficult to classify correctly, especially in cases, like this one, where the classifiers are trained at a different time with respect to the classified points. A possible explanation for this particular mis-classification comes from Figure 1. There, we notice that the plasma density and pressure in the Sunwards inner magnetospheric regions have values compatible with those of certain inner magnetosheath regions. This may have pulled

nodes mapping to the two regions close in the SOM, and in fact we see that cluster 3 and 5 are neighbors in the feature maps of Figure 5.

        In Figure 6, panel g to i, we explore classification results in the case of a mixed-time ("time-variable", TV) training data set, in the meridional plane only.

        In our visualization procedure, the cluster number (and hence the cluster color) is arbitrarily assigned. Hence, clusters

mapping to the same magnetospheric regions may have different colors in classification experiments with different feature sets. For easier reading, we match a posteriori the cluster colors in the different classification experiments to those in Figure 4.

        Contrary to what shown before, now we train our map with points from three different simulated times, $t_0 + 125, 175,$ 200 minutes. The features used are the F1 feature set and the map hyper-parameters are $q = 10 \times 12$, $\epsilon_0 = 0.25$, $\sigma_0 = 1$. The classified data points are a validation set from t=$t_0 + 150, 210, 225$ minutes. We see that the classification results agree quite

well with those shown in Figure 6, panel a to c. One minor difference is the fact that the two magnetosheath clusters just downstream the box shock do not change significantly with time in panels d to f, while they did so in panel a to c. Another difference can be observed in the inner magnetospheric region. In panel c, inner magnetospheric plasma at $t_0 + 225$ was

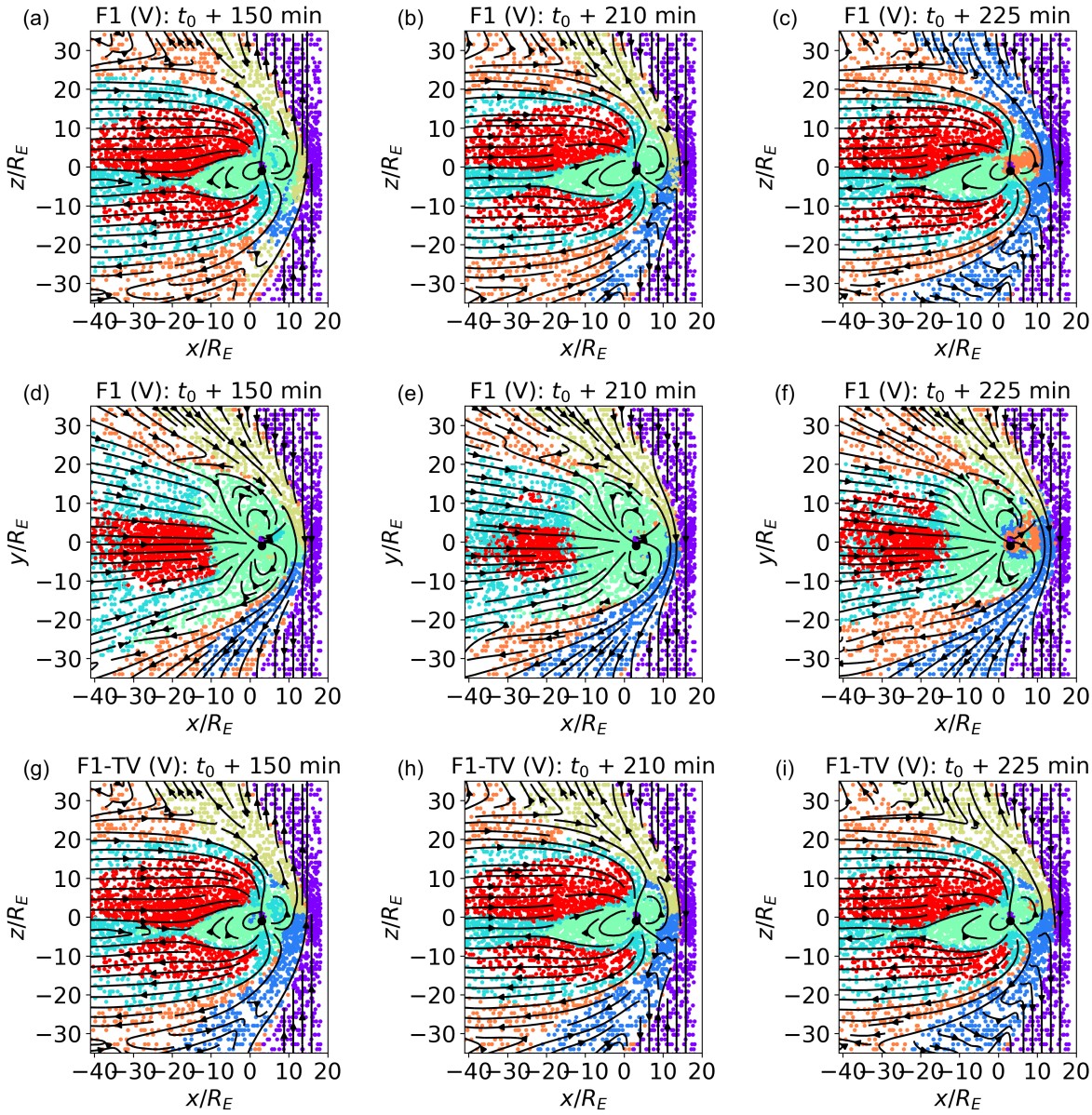

**Figure 6.** Classification of validation ("V") data sets in the meridional (a to c) and equatorial (d to f) planes at $t_0 + 150$, $t_0 + 210$, $t_0 + 225$ minutes. In panel a to f, the points are classified with the same classifiers as Figure 4; data points in panels b and e are from the same simulated time as the training set, in panels a, c, d and f from different times. $B_z$ points Northwards at $t_0 + 150$ minutes, Southwards at $t_0 + 210$, 225 minutes. In panel g to i, the classifiers are trained with a mixed-time data set composed of points from $t_0 + 125$, 175, 200 minutes, feature set F1-TV.

mis-classified as magnetosheath plasma. In panels g, inner magnetospheric plasma at $t_0$+ 150 minutes is mis-classified as boundary layer plasma, possibly a less severe mis-classification. In panel i, the number of mis-classified points in the inner magnetosphere is negligible with respect to panel c.

### 4.2.2  Comparison with different unsupervised classification methods

Another form of model validation consists in comparing classification results with those from another unsupervised classification model. Here, we compare with pure K-means classification.

In Figure 7 we present in panel a, b and c the classification results for the $x/R_E = 0$, $y/R_E = 0$ and $z/R_E = 0$ planes at time $t_0 + 210$ minutes (panel b and c are reproduced here from Figure 4 for ease of reading). We contrast them in panels d, e, and f with K-means classification, also with $k = 7$, using the same features and in the same planes.

Comparing panel a and d, b and e, c and f, we see that the two classification methods give quite similar results. We had remarked, in Figure 4, on the fact that few points, possibly located inside the bow shock, are classified as inner magnetosheath plasma, orange. We notice that the same happens in panels d, e, f.

One difference between the two methods is visible in panel b and e: the magnetosheath cluster associated to the North sector extends a few points further Southwards in panel e than in panel b. This is a minimal difference that can be explained with the fact that the two clusters map to very similar plasma, as remarked previously.

A rather more significant difference can be seen comparing panel c and f. In panel f, some plasma regions rather close to Earth and deep into the inner magnetosphere are classified as magnetosheath plasma (brown) rather than as inner magnetospheric plasma (green). In panel c, they are classified as the latter (green), a classification that appears more appropriate for the region, given its position.

To compare the two classification methods quantitatively, we calculate the number of points which are classified in the same cluster with SOMs plus K-means vs pure K-means classification. 92.15 % of the points are classified in the same cluster, 92.74 % if the two magnetosheath clusters just downstream the bow shock are considered the same. These percentage are calculated on the entire training dataset at time $t_0 + 210$ minutes, of which cuts are depicted in the panels in Figure 7.

We therefore conclude that the classification of SOM nodes and "simple" K-means classification globally agree. An advantage of using SOM with respect to K-means is that the former reduces mis-classification of a section of inner magnetospheric plasma, the region most challenging to classify correctly. Furthermore, SOM feature maps give a better representation of feature variability within each cluster than K-means centroids. This representation can be used to assess feature variability within the cluster. In K-means, only the feature values at the centroid (meaning, one value per class) are available.

### 4.3  On the choice of training features

Up to now, we have used as features for SOM training the three components of the magnetic field and of the velocity and the logarithms of the plasma density, pressure, temperature. We label this feature set F1 in Table 1, where we list several other feature sets we experiment with. In this Section, we show classification results for different feature sets, listed in Table 1, and we aim at obtaining some insights into what constitutes a "good" set of features for our classification purposes.

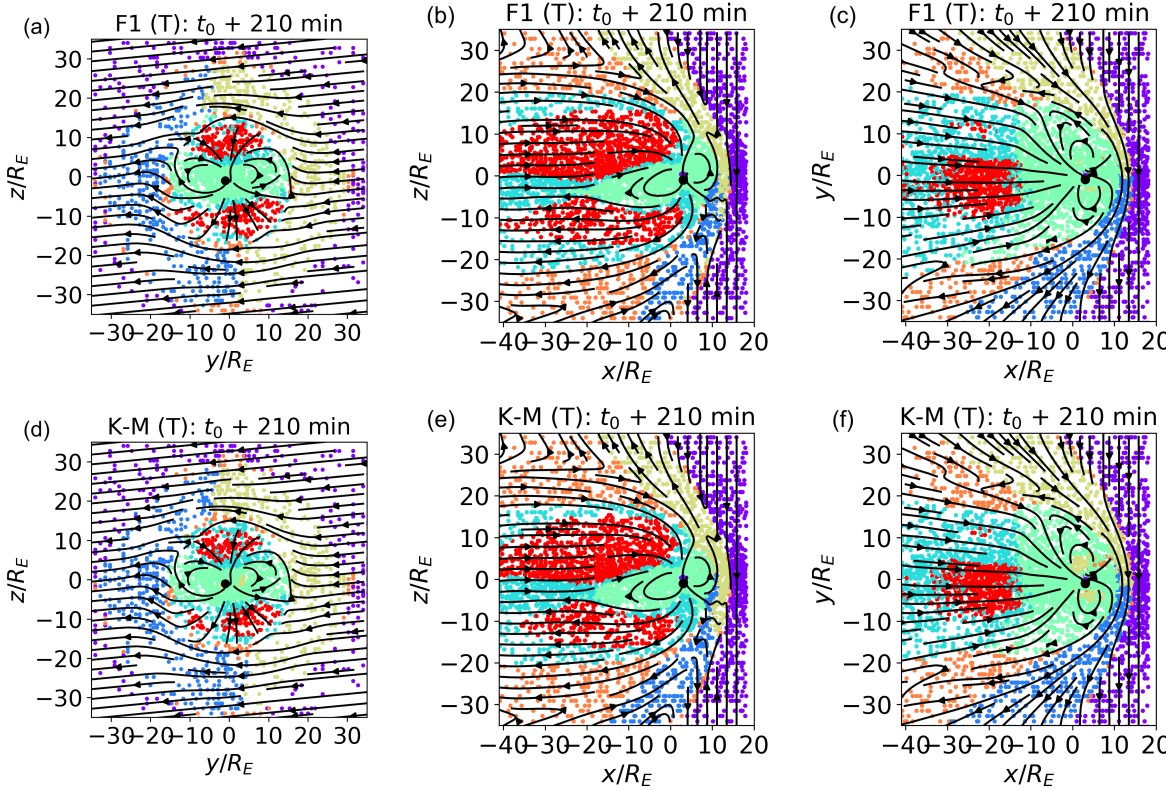

**Figure 7.** Unsupervised K-means classification of trained SOM nodes, with $k = 7$ (panel a, b, c), and pure K-means classification, with $k = 7$ (panel d, e, f). The feature set is F1, the time is $t_0 + 210$ minutes. The training (T) dataset is depicted.

The SOM hyper-parameters are the same in all cases, $q = 10 \times 12$, $\sigma_0 = 1$, $\epsilon_0 = 0.25$. In all cases, $k = 7$. The data used for the training are from $t_0 + 210$ minutes. In Figure 8, 9 and 10 validation (V) datasets are depicted.

In Figure 8, panel a to c, we show sub-standard classification results obtained with non-optimal feature sets. In panel a, F1-NL ("Not-Logarithm") uses density, pressure, temperature, rather than their logarithms. In panel b, F2, we eliminate from the feature list the logarithm of the plasma density, i.e. the most relevant feature for the calculation of the PCs for F1. In panel c, F3, we do not use the Sun-Earth velocity. We see that F1-NL groups together magnetosheath and solar wind plasma (probably the biggest possible classification error), and inner magnetospheric regions are not as clearly separated as in F1. F2 mixes inner magnetospheric and boundary layer data points (green and cyan), and magnetosheath regions just downstream the bow shock and internal magnetosheath regions (orange). With F3, some inner magnetospheric plasma is classified as magnetosheath plasma, already at $t_0 + 210$ minutes.

When analysing satellite data, variables such as the Alfvén speed $v_A$, the Alfvén Mach number $M_A$, the plasma beta $\beta$ provide precious information on the state of the plasma. In Figure 8, panel d to f, we show training set results for the F4, F5 and F6 feature sets, where we add to our "usual" feature list, F1, the logarithm of $v_A$, $M_A$, $\beta$ respectively. Using as features the logarithm of variables, such as $n$, $pp$, $T$, $v_A$, $M_A$, $\beta$, which vary across order of magnitude (see Figure 1), is one of the "lessons learnt" from Figure 8, panel a. Comparing Figure 8, panel d to f, with Figure 4, we see that introducing $log(v_A)$ in the feature list slightly alters classification results. What we called boundary layer cluster in Figure 4 does not include in F4 points at the boundary between the lobes and the magnetosheath. Perhaps more relevant is the fact that the boundary layer and inner magnetospheric clusters (green and cyan) appear to be less clearly separated than in F1. The classification obtained with F5 substantially agrees with F1. With F6, the boundary layer cluster is slightly modified with respect to F1.

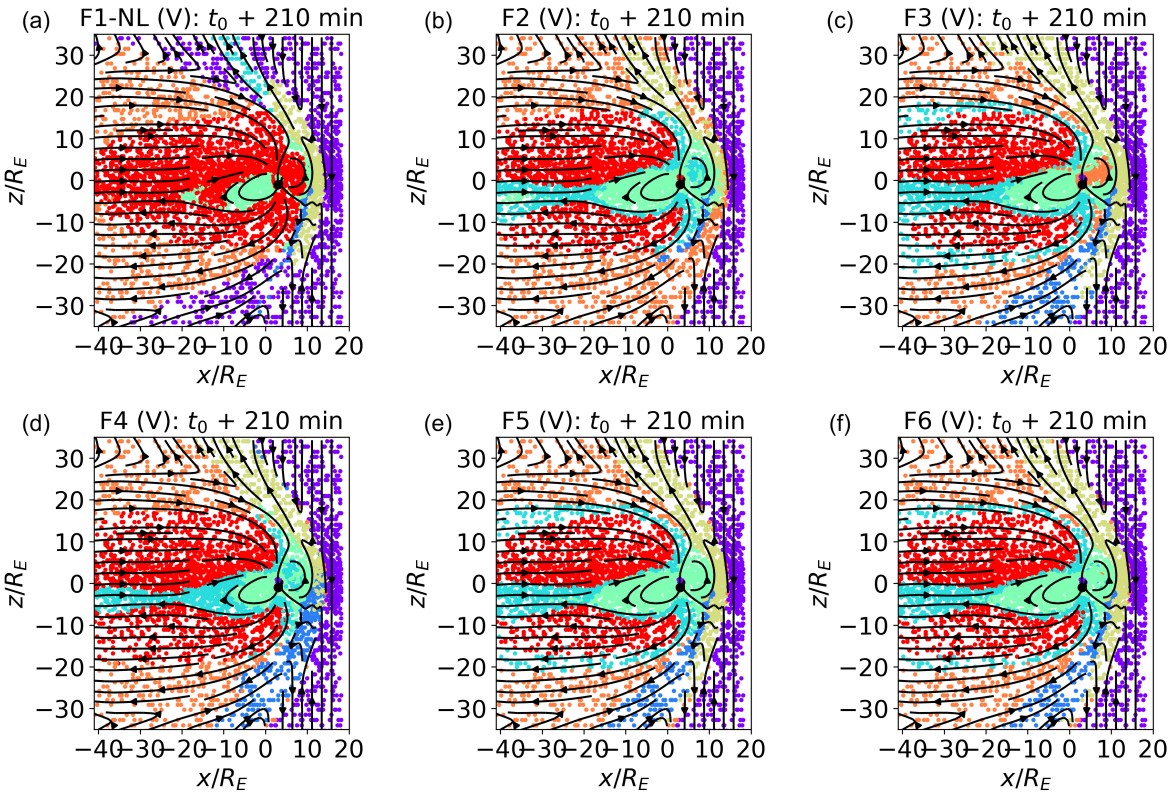

**Figure 8.** Validation plots at $t_0 + 210$ min in the $y/R_E = 0$ plane, from SOMs trained with feature sets F1-NL, F2, F3, F4, F5, F6. F1-NL uses density, pressure, temperature, rather than their logarithms. F2 does not include $log(n)$, F3 does not include $v_x$. F4, F5, F6 add to the feature set F1 the logarithms of Alfvén speed $v_A$, the Alfvénic Mach number $M_A$ and the plasma beta $\beta$, respectively.

**Table 3.** Percentage of data points classified in the same cluster as F1, for the different feature sets (second column, header "S"). In the third column, header "M", the two magnetosheath clusters just downstream the bow shock, 1 and 4, are considered one. The dataset used is the validation dataset, at time $t_0 + 210$ minutes.

| Case | S | M |
|------|------|------|
| F1-TV | 80.71 | 85.72 |
| F1-NL | 59.83 | 61.47 |
| F2 | 84.69 | 84.71 |
| F3 | 87.85 | 89.01 |
| F4 | 82.70 | 83.01 |
| F5 | 93.07 | 94.42 |
| F6 | 91.02 | 92.36 |
| F7 | 83.38 | 84.77 |
| F8 | 91.55 | 91.78 |
| F9 | 66.05 | 75.67 |
| F10 | 92.49 | 93.90 |

In Table 3, second column ("S"), we report the percentage of data points classified in the same cluster as F1 for each of the feature sets of Table 1, for the validation dataset at $t_0 + 210$ minutes. In the third column ("M"), we consider cluster 1 and 4 as a single cluster: in the previous analysis, we remarked that cluster 1 and 4 (the two magnetosheath clusters just downstream the bow shock) map to the same kind of plasma. We keep this into account when comparing classification results with F1. The metrics depicted in Table 3 cannot be used to assess the quality of the classification per se, since we are not comparing against

ground truth, but merely against another classification experiments. However, it gives us a quantitative measure of how much different classification experiments agree.

Comparing Figure 8 with Table 3 results we see, as one could expect, that sub-standard feature sets (F1-NL, F2, F3) agree less with F1 classification than F4, F5, F6. This is the case with F1-NL in particular, which exhibit the lower percentage of similarly classified points with respect to F1. We see that the agreement is particularly good with F5, as already noticed in

Figure 8.

When discussing Table 2, we remarked on the seemingly negligible role that the magnetic field components appear to have in determining the first three PCs, both when their values are not clipped ("F1 feature set") and when they are ("F8 feature set - B clipped"). Here, we investigate if this reflects in classification results.

In Figure 9 we show classification of validation data sets at time $t_0 + 150, 210, 225$ minutes for the F7 feature set (panel a

to c), which do not include the magnetic field, and for F8 (panel d to f), where the magnetic field components are present but clipped as described in Section 2.

Comparing Figure 9, panel a to f, with Figure 6 (the validation plot for F1), we see that the identified clusters are indeed rather similar, including the variation with time of the outer magnetosheath clusters, see discussion of Figure 6. The main

difference with Figure 6 is the fact that, in the F7 case, the boundary layer cluster does not include most of the data points at the boundary between the lobe and magnetosheath plasma (this reflects in the percentage of similarly classified points in Table 3). The boundary layer cluster for F8 instead corresponds quite well with F1. As already observed in Figure 6, inner magnetospheric plasma is the most prone to mis-classification in the validation test. In the case without magnetic field, F7, as also in F1, the mis-classified inner magnetospheric plasma is assigned to the inner magnetosheath cluster. In F8, it is assigned to either boundary layer plasma, at time $t_0+ 150$ minutes, or to one of the magnetosheath clusters just downstream the bowshock, at $t_0 + 225$ minutes.

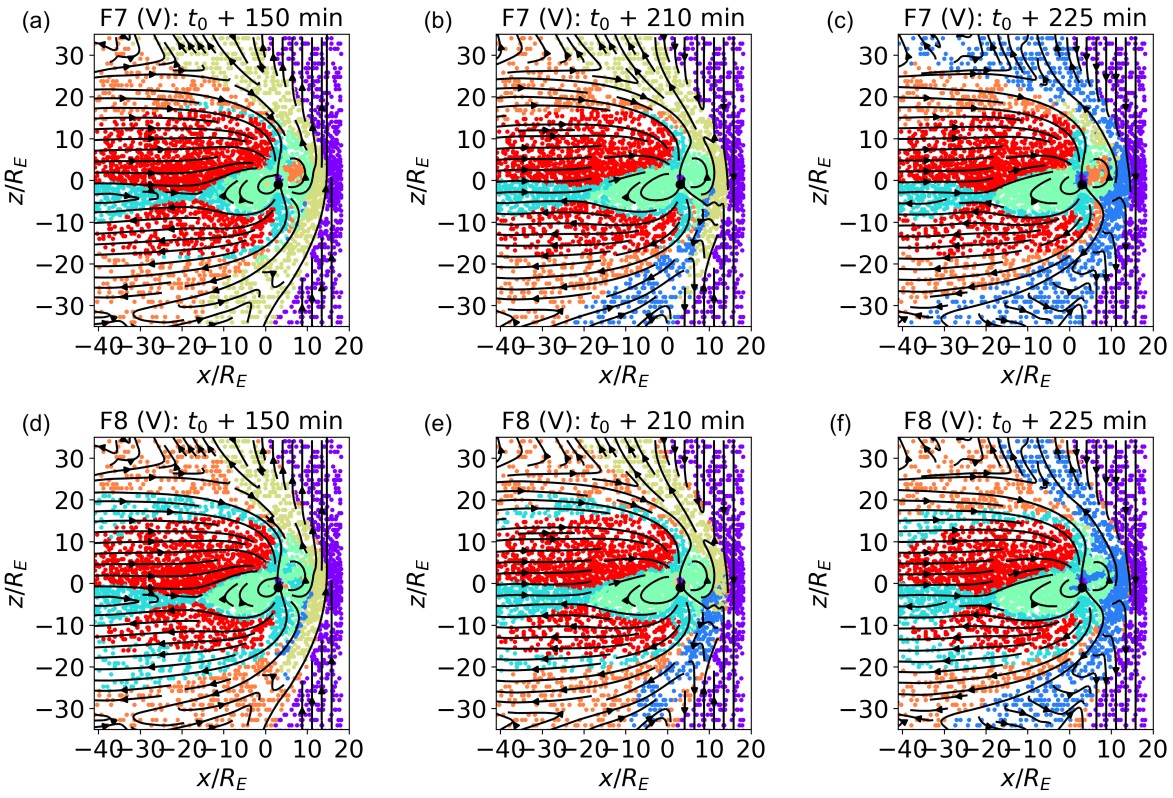

**Figure 9.** Classification of validation data sets: $y/R_E = 0$ plane at $t_0+$ 150, 210, 225 minutes, for maps trained with the F7 and F8 feature sets. In F7, $B_x$, $B_y$, $B_z$ are not used for the map training. In F8, $B_x$, $B_y$, $B_z$ are clipped as described in Section 4.

In Figure 10, panel a to c, we plot the classification results for the validation data sets for a map trained with feature set F9, where instead of the three components of the magnetic field we use only its magnitude, clipped as described above. We see that the green and blue clusters correspond to the regions of highest magnetic field, closer to the Earth. The green and blue regions map to high-$|B|$ regions of the inner magnetosphere and lobes respectively. We notice that with this choice of

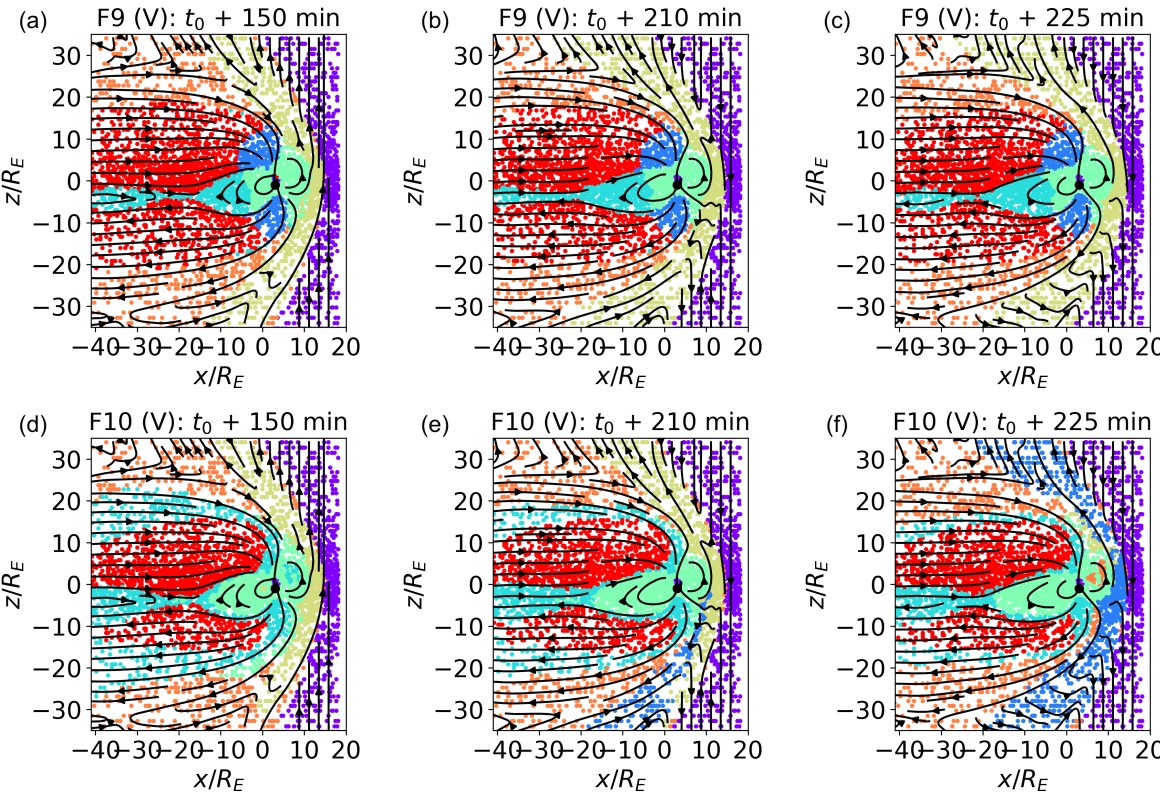

**Figure 10.** Classification of validation data sets: $y/R_E = 0$ plane at $t_0 +$ 150, 210, 225 minutes, for maps trained with the F9 and F10 feature sets. In F9 and F10, the module of the magnetic field, instead of its components, is used. In F9, $|B|$ is clipped as described in Section 4. In F10, the logarithm of the module of the magnetic field is used.

features the inner magnetospheric is consistently classified as such at different simulated times, contrarily to what happened with feature sets previously discussed. The remaining inner magnetospheric plasma is classified together with the current sheet in the cyan cluster (while in F1 inner magnetosphere and current sheets were clearly separated), which does not include plasma at the boundary between magnetosheath and lobes. Magnetosheath plasma just downstream the bow shock is now classified in a single cluster. This classification is consistent with our knowledge of the magnetosphere, and is very robust to temporal

variation. However, it differs significantly from F1 classification, hence the low percentage of similarly classified points in Figure 3.

     F10, depicted in panels d to f, looks remarkably similar to F1 (see also Table 3), especially in the internal regions, including the mis-classification of some inner magnetospheric points as magnetosheath plasma at $t_0 + 225$ minutes. The three magnetosheath cluster vary at the three different times depicted with respect to F1. This behavior, and the pattern of classification

of magnetosheath plasma in F9, shows that the magnetic field is a feature of relevance in classification especially for magnetosheath regions. F10 classification results show that the blue cluster in F9 originates from the clipping procedure. This somehow artificial procedure is however beneficial for inner magnetospheric points, which are not mis-classified in that case.

From this analysis, we learn important lessons on possible different outcomes of the classification procedure and on how to choose features for SOM training.

First of all, we can divide our feature sets into acceptable and sub-standard. Sub-standard feature sets are those, such as F1-NL and F2, that fail to separate plasma regions characterized by highly different plasma parameters. Examining the feature list in F1-NL, the reason for this is obvious: as one can see at a glance from Figure 1, panel g to l, only using a logarithmic representation allows to appreciate how features that span orders of magnitude vary across magnetospheric regions. The "lesson learned" here is to use the data representation that more naturally highlights differences in the training data.

In F2, we excluded from the feature list $log(n)$, which expresses a large percentage of the variance of the training set, with poor classification results: a good rule of thumb is to always include this kind of variables into the training set.

With the exception of F1-NL and F2, all feature sets produce classification results which are first of all quite similar, and generally reflect well our knowledge of the magnetosphere. Differences arise with the inclusion of extra variables, such as in Figure 8, panel d to f, where we add $log(v_A)$, $log(M_A)$ and $log(\beta)$ to F1. All these quantities are derived from base simulation quantities and, while quite useful to the human scientist, do not seem to improve classification results here. In fact, occasionally they appear to degrade them, at least in panel d. These preliminary results therefore point in the direction of not including somehow duplicated information into the training set. One might even argue that the algorithm is "smart" enough to "see" such derived variables as long as the underlying variables are given.

Some feature sets, F1 (Figure 4 and 6) and F9 (Figure 10), raise particular interest. Other feature sets, such as F5, F6, F10, albeit interesting in their own right, essentially reproduce the results of F1. Both F1 and F9 produce classification results which, albeit somehow different, separate well known magnetospheric regions. In F9, less information than in F1 is made available to the SOM: we use magnetic field magnitude rather than magnetic field components for the training. This results into two clusters, green and blue, that clearly correspond to high-$|B|$ points, whose values have been clipped as described above. This classification appears more robust to temporal variation than F1, perhaps because all three PCs (and not just the first two, as for F1) present well-defined multi-peaked distributions. This confirms the well-known fact that multi-peaked distributions of the input data are a very relevant factor in determining classification results.

We remark that these insights have to be further tested against different classification problems, and may be somehow dependent on the classification procedure we chose in our work.

## 5   Conclusions

The growing amount of data produced by magnetospheric missions is amenable to the application of ML classification methods, that could help clustering the hundreds of gigabits of data produced every day by missions such as MMS into a small number of clusters characterized by similar plasma properties. Argall et al. (2020), for example, argue that ML models could be used

to analyze magnetospheric satellite measurements in steps: first, region classifiers would separate between macro-regions, as the model we propose here does. Then, specialized event classifiers would target local, region-specific, processes.

Most of the classification works focusing on the magnetosphere consist of supervised classification methods. In this paper, instead, we present an *unsupervised* classification procedure for large scale magnetospheric regions based on Amaya et al. (2020), where 14 years of ACE solar wind measurements are classified with a techniques based on SOMs. We choose an unsupervised classification method to avoid relying on a labelled training set, which risks introducing the bias of the labelling scientists into the classification procedure .

As a first step towards the application of this methodology to spacecraft data, we verify its performance on simulated magnetospheric data points obtained with the MHD code OpenGGCM-CTIM-RCM. We choose to start with simulated data since they offer several distinct advantages. First of all, we can for the moment bypass issues, such as instrument noise and instrument limitations, that are unavoidable with spacecraft data. Data analysis, de-noising, pre-processing is a fundamental component of ML activities. With simulations, we have access to data from a controlled environment that need minimal pre-processing, and allow us to focus on the ML algorithm for the time being. Furthermore, the time/ space ambiguity that characterizes spacecraft data is not present in simulations, and it is relatively easy to qualitatively verify classification performance by plotting the classified data in the simulated space. Performance validation can be an issue for magnetospheric unsupervised models working on spacecraft data. A model such as ours, trained and validated against simulated data points, could be part of an array of tests against which unsupervised classifications of magnetospheric data could be benchmarked.

The code we are using to produce the simulation is MHD. This means that kinetic processes are not included in our work, and that variables available in observations, such as parallel and perpendicular temperatures and pressures, moments separated by species, are not available to us at this stage. This is certainly a limitation of our current analysis. This limitation is somehow mitigated by the fact that we are focusing on classification on large scale regions. Future work, on kinetic simulations and spacecraft data, will assess the impact of including "kinetic" variables among the classification features.

We obtain classification results, e.g. Figure 4 and Figure 10, that match surprisingly well our knowledge of the terrestrial magnetosphere, accumulated in decades of observations and scientific investigation. The analysis of the SOM feature maps, Figure 5, shows that the SOM node values associated with the different features represent well the feature variability across the magnetosphere, at least for the features that contribute most to determine the principal components used for the SOM training. Roaming across the feature map, we get hints of the processes characterizing the different clusters, see the discussion on plasma compression and heating across the bow shock.

Our validation analysis in Section 4.2 shows that the classification procedure is quite robust to temporal evolution in the magnetosphere. In particular, consistent results are produced with opposite orientation of the $B_z$ IMF component, which has profound consequence for the magnetospheric configuration.

Since this work is intended as a starting point rather than as a concluded analysis, we report in details on our exploration activities in terms of SOM hyper-parameters (A) and feature sets (Section 4.3). We hope that this work will constitute a useful reference for colleagues working on similar issues in the future. In Section 4.3, in particular, we highlight our "lessons learned" when exploring classification with different feature sets. They can be summarized as follows: a) the most efficient features

are characterized by multi-peaked distributions; b) when feature values are spread over orders of magnitude, a logarithmic representation is preferable; c) a preliminary analysis on the percent variance expressed by each potential feature can convey useful information on feature selection; d) derived variables do not necessarily improve classification results; e) different choices of features can produce different but equally significant classification results.

In this work, we have focused on the classification of large scale simulated regions. However, this is only one of the classification activities one may want to be able to perform on simulated, or observed, data. Other activities of interest may be the classification of meso-scale structures, such as dipolarizing flux bundles or reconnection exhausts. This seems to be within the purview of the method, assuming that an appropriate number of clusters is used, and that the simulations used to produce the data are resolved enough. To increase the chances of meaningful classification of meso-scale structure, one may consider applying a second round of unsupervised classification on the points classified in the same, large-scale cluster. Another activity of interest could be the identification of points of transition between domains. Such an activity appears challenging in the absence, among the features used for the clustering, of spatial and temporal derivatives. We purposefully refrained from using them among our training features, since we are aiming for a local classification model, that does not rely on higher resolution sampling either in space or time.

Several points are left as future work. It should be investigated whether our classification procedure, while satisfactory at this stage, could be improved. Possible venues of improvement could be the use of a dimensionality reduction technique that does not rely on linear correlation between the features, or the use of dynamic (Rougier and Boniface, 2011b) rather than static SOMs. As an example, in Amaya et al. (2020), more advanced pre-processing techniques were experimented with, which will most probably prove useful when we will move to the more challenging environment of spacecraft observations (as opposed to simulations). Furthermore, Amaya et al. (2020) employed windows of time in the classification, which we have not used in this work in favor of an "instantaneous" approach. In future work, we intend to verify which approach gives better results.

It should also be verified whether similar modifications reduce the mis-classification of inner magnetospheric points observed with a number of feature sets including F1, and if they reduce the importance or outright eliminate the need of looking for optimal sets of training features.

The natural next step of our work is the classification of spacecraft data. There, many more variables not included in an MHD descriptionwill be available. They will probably constitute both a challenge and an opportunity for unsupervised classification methods, and will allow to attempt classification aimed at smaller scale structure, where such variables are expected to be essential. Such procedures will be aimed not at competing in accuracy with supervised classifications, but they will hopefully be pivotal in highlighting new processes.

**Appendix A: Exploring the SOM hyper-parameters: node number, initial lattice neighbor width, learning rate**

SOMs are characterized by several hyper-parameters (Section 3): the number of rows $Lr$ and of columns $Lc$, the initial learning rate $\epsilon_0$, the initial lattice neighbor width $\sigma_0$. In this Section, we explore how changing the hyper-parameters changes the convergence of the map. The features we use are F1 in Table 1. Libraries for the automatic selection of SOM hyper-parameters

are available; however we prefer, at this stage, manual hyper-parameter selection, to familiarize ourselves with the classification procedure and expected outcomes in different simulated scenarios.

In Figure A1 we see the evolution of the quantization error $Q_e$, Eq. 4, with the number of iterations, $\tau$, changing the number of nodes (panel a), the initial learning rate $\epsilon_0$ (panel b), the initial lattice neighbor width $\sigma_0$ (panel c). The number of iterations used is larger than three epochs, to ensure that each training data point is presented to the SOM an adequate number of times. In Figure A1, panel a to c, the standard deviation of the quantization errors strongly reduces as a function of the iteration number, a consequence of the iteration number-dependent evolution we impose on the learning rate and on the lattice neighbor width.

A more recent version of the SOM that does not depend on the iteration number has been proposed by Rougier and Boniface (2011a). This Dynamic Self-Organizing Map (DSOM) has been succesfully used by Amaya et al. (2020) to classify different solar wind types. In this work, we have decided to use the original SOM algorithm as the results already show very good convergence to meaningful classes.

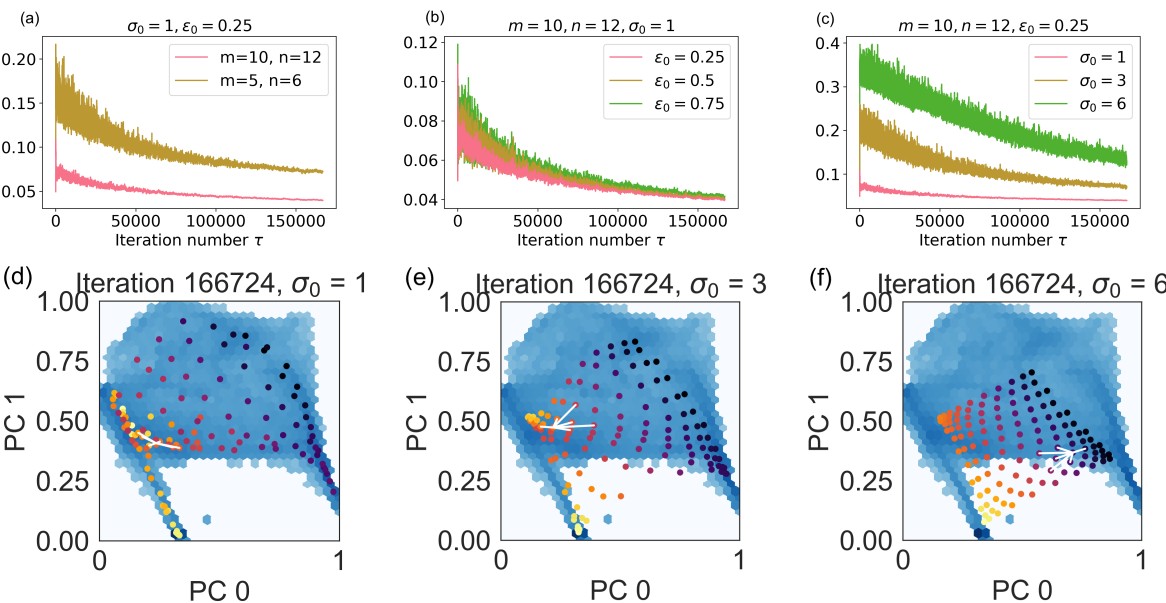

**Figure A1.** Hyper-parameter selection: in panels a to c, evolution of the quantization error $Q_E$ with the number of iterations $\tau$, changing the SOM node number ($q$, panel a), initial learning rate ($\epsilon_0$, panel b), initial lattice neighbor width ($\sigma_0$, panel c). In panels d to f, code words 0 and 1 associated with SOM nodes (in bright colors) are superimposed to the input point distribution (blue background) for a SOM with $q = 10 \times 12$ nodes, initial learning rate $\epsilon_0 = 0.25$ and initial lattice neighbor width $\sigma_0$=1 (panel d), 3 (panel e), 6 (panel f), at the final iteration. In the animation, we show the evolution of the node positions with the iteration number. The white lines connect node (3,3) to its nearest neighbors.

In Figure A1, panel a, we observe that the quantization error decreases with increasing number of nodes. This is not surprising, since the quantization error measures the average distance between each input data point and its BMU. With a larger number of nodes, this distance naturally decreases. In panel b, we see that decreasing the initial learning rate $\epsilon_0$ does not change significantly the average error value. However, smaller oscillations around the average value are observed with lower learning rates. In panel c, we observe that changing the initial lattice neighbor width $\sigma_0$ has a significant impact on the quantization error. The reason is clear when looking at panel d to f and respective animations.

In panels d to f we depict as colored dots the code words associated with the SOM nodes obtained with $\sigma_0$=1 (panel d), 3 (panel e) and 6 (panel f): each of the brightly colored dots corresponds to one of the $\mathbf{w}_i$, with $0 \leq i < q$, and $q = 10 \times 12$, defined in Section 3. The code words are depicted in the reduced space obtained after dimensionality reduction of the original features with PCA. Of the three principal components, PCs, that characterize the reduced space, we show here only the PC0 vs PC1 distribution. The darker, continuous background shows the distribution of the code words associated with the input points (i.e., the $\mathbf{x}_\tau$ in Section 3), again for PC 0 and 1. We see that in panel d the nodes (i.e., the dots) superimpose well the data distribution. In particular, higher node density is observed in correspondence of the darker areas of the underlying distribution, where data point density is higher. There are no SOM nodes in the "white" area, where data points are not present. With larger $\sigma_0$'s, panel e and f, we observe that the node distribution maps the data distribution less optimally. Larger lattice widths mean that a single new data point significantly affects a larger number of SOM nodes. High $\sigma_0$'s, then, "drag" a large number of map nodes closer to the location in the PC0 vs PC1 plot of every new data point. We see this in the animation of panels d to f of A1, where the SOM nodes move across the PC0 vs PC1 plane as a function of the iteration number $\tau$, for the different lattice neighbor width values.

At the end of this manual hyper-parameter investigation, we choose $q = 10 \times 12$, $\epsilon_0 = 0.25$, $\sigma_0 = 1$ for our maps.

**Appendix B: Classification evolution with the cluster number**

In this Appendix, we explore how the classification of magnetospheric regions changes when the number of K-means clusters $k$ used to classify the SOM nodes is reduced. In all the cases described here (as in the rest of the paper, unless otherwise specified), the SOM map is obtained with the F1 features in Table 1 and with $q = 10 \times 12$, $\epsilon_0 = 0.25$, $\sigma_0 = 1$. The $k = 7$ case is further described in Section 4.1.

In Figure B1, we show the classification results with $k = 6$ (panel a, b) and $k = 5$ (panel c and d). The data depicted are the training dataset. The $k = 7$ case is depicted in Figure 4. In panel a, c and b, d we depict the simulated meridional and equatorial plane, with the data points colored according to their respective clusters. Following the changes from $k = 7$ to $k = 5$ allows us better insights into our classification procedure, and also highlights which of the magnetospheric regions are more similar in terms of plasma parameters.

In panel a and b we plot $k = 6$. Comparing them with Figure 4, we see that reducing the number of clusters of one unit merges the three magnetosheath clusters, brown, orange and blue, into two, blue and orange. This is consistent with the fact

that the magnetosheath clusters, and in particular cluster 4 (brown) and 1 (blue) with $k = 7$, map to quite similar plasmas. The more "internal" clusters (inner magnetosphere, boundary layers, lobes) are not affected.

Further reducing to number of clusters to $k = 5$, instead, affects mostly the "internal" clusters. The boundary layer cluster disappears, and the points that mapped to it are, quite sensibly, assigned to the clusters mapping to inner magnetospheric (this is mostly the case of current sheet plasma), magnetosheath or lobe plasma (the points at the boundary between lobes and magnetosheath). Some inner magnetosphere points are mis-classified as magnetosheath plasma. With both $k = 6$ and $k = 5$ the solar wind cluster, that differs the most from the others, is left unaltered.

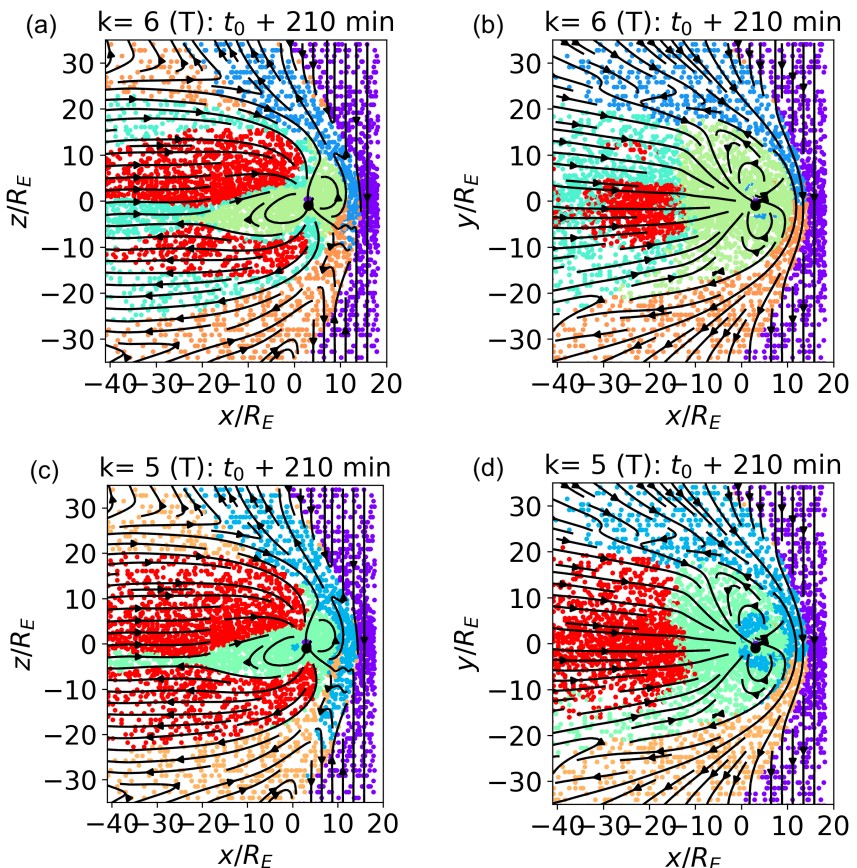

**Figure B1.** $y/R_E = 0$ (panel a and c) and $z/R_E = 0$ (panel b and d) cuts with $k = 6$ (panel a, b) and $k = 5$ (panel c, d). The training dataset is depicted.

In Figure B2, we depict the pure K-means classification with $k = 6$ (panel and b) and $k = 5$ (panel c and d) for data points in the meridional (panel a and c) and equatorial (panel b and d) planes, to be compared with the SOM classification depicted in

Figure B1. We do not notice any significant difference between the SOM and K-means classification for $k = 5$; for $k = 6$ we notice, as already with $k = 7$, that the SOM classification reduces the mis-classification of internal magnetospheric points.

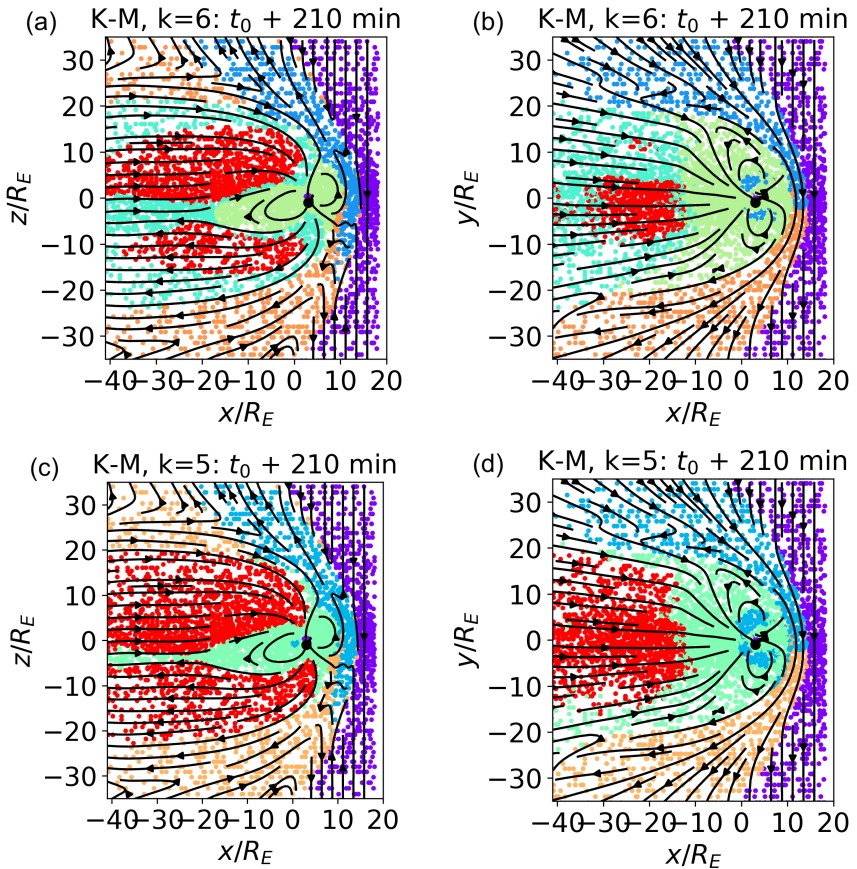

**Figure B2.** K-means classification with $k = 6$ (panel and b) and $k = 5$ (panel c and d) for the data points in the meridional (panel a and c) and equatorial (panel b and d) planes. The training dataset is depicted.

In summary, decreasing $k$ from a larger to a smaller number produces a more coarse-grained classification. Generally speaking, every time $k$ is decreased, the three clusters mapping to the most similar plasma reorganize and coalesce into two. This process shows which magnetospheric regions are most similar.

*Author contributions.* M.E.I. run the simulation and performed the analysis. J.A. assisted in the analysis. J.R. and B.F. provided the OpenGGCM-CTIM-RCM code and support in using it. R.D. and G.L. supported the investigation and provided useful advice.

*Competing interests.*   No competing interests are present

*Code availability.*   OpenGGCM-CTIM-RCM is available at the Community Coordinated Modeling Center at NASA/GSFC for model runs on demand (see: http://ccmc.gsfc.nasa.gov).

*Data availability.*   The simulation data set ("KUL_OpenGGCM") is available from Cineca AIDA-DB, the simulation repository associated with the H2020 AIDA project. In order to access the meta-information and the link to "KUL_OpenGGCM" simulation, please refer to the tutorial at http://aida-space.eu/AIDAdb-iRODS.

*Acknowledgements.*   We acknowledge funding from the European Union's Horizon 2020 research and innovation programme under grant agreement No 776262 (AIDA, Artificial Intelligence for Data Analysis, www.aida-space.eu). Work at UNH was also supported by grant AGS-1603021 from the National Science Foundation and by award FA9550-18-1-0483 from the Air Force Office of Scientific Research. The OpenGGCM-CTIM-RCM simulations were performed on the supercomputer Marconi-Broadwell (Cineca, Italy) under a PRACE allocation. We acknowledge the use of the MiniSom (Vettigli), scikit-learn (Pedregosa et al., 2011), pandas, matplotlib python packages.

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
