# Peer review of "Unsupervised classification of simulated magnetospheric regions"

_Annales Geophysicae, 2021_

## Author Comment (AC2)

**Table 3.** Percentage of data points classified in the same cluster as F1, for the different feature sets (second column, header "S"). In the third column, header "M", the two magnetosheath clusters just downstream the bow shock, 1 and 4, are considered one. The dataset used is the validation dataset, at time $t_0 + 210$ minutes.

| Case | S | M |
|------|-------|-------|
| F1-TV | 80.71 | 85.72 |
| F1-NL | 59.83 | 61.47 |
| F2 | 84.69 | 84.71 |
| F3 | 87.85 | 89.01 |
| F4 | 82.70 | 83.01 |
| F5 | 93.07 | 94.42 |
| F6 | 91.02 | 92.36 |
| F7 | 83.38 | 84.77 |
| F8 | 91.55 | 91.78 |
| F9 | 66.05 | 75.67 |
| F10 | 92.49 | 93.90 |

---

## Author Response (AR1)

**Response to the reviewers**

We thank the reviewers for their feedback on our work. We respond to their comments point by point, elaborating on the responses already provided during the open discussion. We want to remark that the response to reviewer 2 in particular has been significantly expanded with respect to the open discussion.

Modifications are marked in red in the revised manuscript.

As requested by both referees, we have matched the colors of the clusters in the different classification experiments to those of the F1 feature set, to improve readability. Furthermore, we have added quantitative estimates of the agreement of classification results with different sets of features.

As requested by the editor, we have expanded the chapter describing the SOMs, to make it clearer for readers from the space physics community.

**Referee 1**

Anonymous review of Unsupervised classification of simulated magnetospheric regions by Innocenti et al for Annales Geophysicae

In the manuscript under review, the authors have applied machine learning, more specifically self-organizing maps (SOMs), to the question of identifying the magnetospheric regions in which space plasma measurements were made. They propose that their method could be applied to deciding which parts of large measurement databases be downlinked at high resolution, a very real issue for space missions. In this study, they apply the SOM method and K-means clustering to simulation data generated by the OpenGGCM-CTIM-RCM code, a global MHD simulation of the Earth's magnetosphere. They show the method is capable of successfully classifying several magnetospheric plasma regions, and perform comparisons of input data preparation on classification results.

The manuscript is well written, describes new science results, and is of high interest to the community. The application does not contain brilliant breakthroughs as such, but is a valuable addition and a useful parameter study. There are only a few clarifications and improvements to the discussion to suggest before recommending publication, which I have divided below into major and minor suggestions.

Major points:

L94-95: Please explicitly state that the boundary conditions vary with time. Are all boundary cells surrounding the simulation domain always at the same value, varying identically with time?

The manuscript has been edited as follows:

*The OpenGGCM-CTIM-RCM boundary conditions require the specification of the three components of the solar wind velocity and magnetic field, the plasma pressure and the plasma number density at 1 AU. Boundary conditions in the sunward direction vary with time. They are interpolated to the appropriate simulated time from ACE observations (Stone et al., 1998), and applied identically to the entire sunward boundary. At the other boundaries, open boundary conditions (i.e., zero normal derivatives) are applied, with appropriate corrections to satisfy the $\nabla \cdot \mathbf{B} = 0$ condition.*

L157: Please clarify the data point selection. Is it randomized? Is there any selection based on Y or Z?

We have randomly selected 1 % of the points included in the $-40 < x/R_E < 18$, $-L_y < y/R_E < L_y$, $-L_z < z/R_E < L_z$ subdomain, seeding the random selection to be able to retrieve the same dataset if needed. No selection based on y or z has been used.

This has been clarified in the manuscript as follows:

*The selection of these points is randomized, and the seed of the random number generator is fixed to ensure that results can be reproduced. Tests with different seeds and with an higher number of training points did not result into significantly different classification results.*

Figures and analysis: Although the polar plane is very interesting in many ways and should indeed play a role in the study, the equatorial plane could be considered even more relevant, particularly considering the orbits of recent significant space missions. I strongly urge that you present also equatorial plane plots early on to show the strength of the method.

In the revised version of the manuscript, we have introduced equatorial plane results, previously shown for the first time in Figure 7, in Figure 4, panel e as well. Furthermore, we have added validation plots for the F1 feature set in the equatorial plane in Figure 6, panel d to f, which in the old version of the manuscript showed only the meridional plane.

As commented in the manuscript, equatorial plane results are consistent with the classification in the meridional plane. Also the mis-classifications are consistent: also in the equatorial plane we see mis-classification of a few bow shock points, which are classified as inner magnetosheath (orange), and mis-classification of a few inner magnetosphere points as inner magnetosheath plasma at $t_0 + 225$ minutes (orange).

Table 2: Instead of multiplying values in the latter two feature sets, could all sets perhaps be re-normalized to a norm of 1? This would make comparison easier.

We have removed the multiplication in the first two sets, now the sum of each line is 1.

Kneedle determination of optimal cluster count: did you attempt changing the number of clusters to see how robust the method is? Would setting k=6 merge clusters 1 and 4? How would this change the misclassified cluster 5 points at the bow shock for F1? What about setting k=5, would clusters 1,4 and 5 merge? I believe a written description of such a test without figures would

suffice.

Short answer: $k = 6$ merges the three magnetosheath clusters into two, the other clusters are left unaltered. With $k = 5$, the boundary layer cluster disappears, and the points that mapped to it are assigned to the clusters mapping to inner magnetospheric (this is mostly the case of current sheet plasma), magnetosheath or lobe plasma. In all cases, we keep seeing few bow shock points mis-classified as inner magnetosheath plasma.

We have added in the revised manuscript an Appendix, Appendix B, where we examine classification results with $k = 5$ and $k = 6$.

Could you please add some quantitative estimate of the agreement between different feature sets? This would be particularly useful if you also included comparison against the pure K-clustering approach, to show how much improvement SOMs bring to the table. For this purpose, I think manually merging some clusters (e.g. 1,4,5) would be acceptable.

We have added quantitative estimate of the agreement of the SOMS+ K-means vs pure K-means classification in Section 4.2.2:

*To compare the two classification methods quantitatively, we calculate the number of points which are classified in the same cluster with SOMs plus K-means vs pure K-means classification. 92.15 % of the points are classified in the same cluster, 92.74 % if the two magnetosheath clusters just downstream the bow shock are considered the same. These percentage are calculated on the entire training dataset at time $t_0 + 210$ minutes, of which cuts are depicted in the panels in Figure 7.*

We have added Table 3, where we show, for each of the feature set examined, how many of the points in the validation dataset at time $t_0 + 210$ minutes are classified as in T1 (column S). We also examine the case when the two magnetosheath clusters downstream the bow shock are merger (column M).

L372: Since you talk of lessons learned, I was surprised that you did not describe attempts using the logarithm of magnitude of B, instead going with the quasi-arbitrary clipping procedure. Please try out log(B) if not already attempted, and at least briefly report the results.

We have added in Figure 10, panel d to f, a new feature set, F10, where $log(|B|)$ is used as a feature, instead of clipped $|B|$.

As one can see comparing F1, F9 and F10, F10 looks remarkably similar to F1 in the internal regions, including the mis-classification of some inner magnetospheric points as magnetosheath plasma at $t_0 + 225$ minutes. The three magnetosheath cluster vary at the three different times depicted with respect to F1. This behavior, and the pattern of classification of magnetosheath plasma in F9, shows that the magnetic field is a feature of relevance in classification especially for magnetosheath regions. F10 classification results show that the blue cluster in F9 originates from the clipping procedure. This somehow artificial procedure is however beneficial for inner magnetospheric points, which are not mis-classified in that case.

Conclusions: I would like to see some more discussion about relevance and limitations. An important point to discuss, albeit outside the scope of the current paper, is the identification of

small structures inside larger domains (e.g. the tail current sheet within the boundary layer) and the points of transition between domains. The latter might arise naturally from this method, the former not so much. Some mention of this should be included. Also, there has been no discussion of the drastic difference between MHD simulation descriptions of plasma parameters and the values measured by spacecraft or hybrid simulation methods, namely kinetic effects, noise, and/or instrumentation limitations. Some of these are touched upon in Amaya(2020), but they are quite relevant to the discussion here as well, if this method is indeed to become a first step in any actual classification approach.

We have added this discussion to the Conclusions:

*In this paper, we have classified large scale simulated regions. However this is only one of the classification activities one may want to be able to perform on simulated, or observed, data. Other activities of interest may be the classification of meso-scale structures, such as dipolarizing flux bundles or reconnection exhausts. This seems to be within the purview of the method, assuming that an appropriate number of clusters is used, and that the simulations used to produce the data are resolved enough. To increase the chances of meaningful classification of meso-scale structure, one may consider applying a second round of unsupervised classification on the points classified in the same, large-scale cluster. Another activity of interest could be the identification of points of transition between domains. Such an activity appears challenging in the absence, among the features used for the clustering, of spatial and temporal derivatives. We purposefully refrained from using them among our training features, since we are aiming for a local classification model, that does not rely on higher resolution sampling either in space or time.*

We have edited L420 as follows:

*As a first step towards the application of this methodology to spacecraft data, we verify its performance on simulated magnetospheric data points obtained with the MHD code OpenGGCM-CTIM-RCM. We choose to start with simulated data since they offer several distinct advantages. First of all, we can for the moment bypass issues, such as instrument noise and instrument limitations, that are unavoidable with spacecraft data. Data analysis, de-noising, pre-processing is a fundamental component of ML activities. With simulations, we have access to data from a controlled environment that need minimal pre-processing, and allow us to focus on the ML algorithm for the time being. Furthermore, the time/ space ambiguity that characterizes spacecraft data is not present in simulations, and it is relatively easy to qualitatively verify classification performance by plotting the classified data in the simulated space. Performance validation can be an issue for magnetospheric unsupervised models working on spacecraft data. A model such as ours, trained and validated against simulated data points, could be part of an array of tests against which unsupervised classifications of magnetospheric data could be bench marked.*
*The code we are using to produce the simulation is MHD. This means that kinetic processes are not included in our work, and that variables available in observations, such as parallel and perpendicular temperatures and pressures, moments separated by species, are not available to us at this stage. This is certainly a limitation of our current analysis. This limitation is somehow mitigated by the fact that we are focusing on classification on large scale regions. Future work, on kinetic simulations and spacecraft, will assess the impact of including "kinetic" variables among the classification*

*features.*

Minor points:

Abstract: I would recommend you mention comparisons against the K-clustering event here, as well as the use of PCA to reduce input data.

We have added the following sentences to the abstract:

*The dimensionality of the data is reduced with Principal Component Analysis before classification. ... We validate our classification results by plotting the classified data in the simulated space, and by comparing with K-means classification.*

L14: Introduction to machine learning: Perhaps some more general, canonical paper could be cited?

The sentence has been edited as follows:

*The growing amount of data produced by measurements and simulations of different aspects of the heliospheric environment has made it fertile ground for applications rooted in Artificial Intelligence, AI, and Machine Learning, ML (Bishop, 2006, Goodfellow et al, 2016). The use of ML in space weather nowcasting and forecasting is addressed in particular in Camporeale (2019).*

L30: how has the data magnitude changed when compared with Cluster, the Van Allen probes, ISEE-1, etc?

We have added this sentence to the manuscript:

*The four-spacecraft Cluster mission (Escoubet et al., 2001) has been investigating the Earth's magnetic environment and its interaction with the solar wind for over 20 years. Laakso et al. (2010), introducing a publicly accessible archive for high-resolution Cluster data, expected it to exceed 50 TB.*

L90: Please clarify in better detail the simulation set-up. Is this the domain size of the whole simulation, or the portion of it shown in figure 1? In section 4 you state you use only the relevant portion of the simulation for post-processing SOM analysis. What is the spatial resolution of the simulation?

The description of the simulation set up has been updated as follows:

*OpenGGCM-CTIM-RCM uses a stretched Cartesian grid, which in this work has 325x150x150 cells, sufficient for our large scale classification purposes, while running for few hours on a modest number of cores. The point density increases in the Sunwards direction and in correspondence with the magnetospheric plasma sheet, the "interesting" region of the simulation for our current purposes. The simulation extends from $-3000$ $R_E$ to $18$ $R_E$ in the Earth-Sun direction, from $-36$ $R_E$*

to $+36$ $R_E$ in the y and z direction. $R_E$ is the Earth's mean radius, the Geocentric Solar Equatorial (GSE) coordinate system is used in this study.In this work, we do not classify points from the entire simulated domain. We focus on a subset of the points with coordinates $-41 < x/R_E < 18$, i.e. the magnetosphere / solar wind interaction region and the near magnetotail.

L119: The indicated magnetic field clipping value does not make much sense. Is the intention to keep the same ratio between the three components and the original signs, but re-scale the magnetic field vector to a magnitude of 100 nT? Please rephrase.

This sentence has been added to the manuscript:

The intention of the clipping procedure is to cap the maximum magnitude of the magnetic field module to 100 nT, while retaining information on the sign of each magnetic field component.

The respective ratios of the original field components are lost with this procedure, since each of them is arbitrarily set to $\sqrt{100^2}/3$ $nT$.

L129: is the lattice really of type $R^2$? The visualizations show a hexagonal grid.

The lattice is of type $R^2$, the grid is hexagonal. This is a common choice for SOMs, as described in the review by Tuevo Kohonen, 2014, chapter 3. This sentence has been added to the manuscript:
To clarify the visualization, this sentence will be added to the text:

As it is often done with two-dimensional SOM lattices, the nodes are organized in an hexagonal grid [Kohonen 2014].

L131: I recommend you briefly mention that from available plasma variable you select n features for the SOM method to use, so that the $R^n$ notation is meaningful.

The following text has been added to the manuscript:

n is therefore the number of plasma variables that we select, among the available ones, for our classification experiment.

L135: Could this be better described (to the layman) as altering the feature values of the code word so that the distance ws for the winning element becomes smaller? Similarly, consolidation of terms might make it easier to read, e.g. data entry vs input data point vs input point - these are probably all the same thing, i.e. a list of features associated with a point of measurement.

The comments have been incorporated in the manuscript as follows:

SOMs learn by moving the winning element and neighboring nodes closer to the data entry, based on their relative distance, and on a iteration number-dependent learning rate $\epsilon(\tau)$, with $\tau$ the progression of samples being presented to the map for training: the feature values of the winning element are altered as to reduce the distance between the "updated" winning element and the data entry.

Indeed, data entry, input data and input point are the same thing. We have added a clarification statement in the manuscript:

*Notice that, in the rest of the manuscript, we will use terms such as "data point", "data entry", "input point" interchangeably.*

L143: is the numerator of the exponent in formula (3) the integer lattice neighbor distance, up to a value of sigma(tau)?

$\sigma(\tau)$ is the value of the lattice neighbor width, which is not necessarily an integer. $\sigma(\tau)$ decreases with the iteration number $(\tau)$ to ensure that the map converges at the end of the training.

L201: Is the K-means clustering performed based on the final code words of each node?

Yes.

L282: I would suggest briefly mentioning the sunward inner magnetosphere misclassification already here.

We prefer to complete our line of thought before introducing the inner magnetosphere misclassification, which is mentioned just a few lines below
L286: This should probably be Bx, not Bz?

yes, corrected

L299: Since the cluster numbering is arbitrary, you could perhaps re-order the colors to match the earlier ordering to assist the reader.

The colors of the clusters have been re-ordered to match F1.

Figure 7: What time value is this? (for the caption, not just the text)

$t_0 + 210$ minutes, it has been added in the label.

L337: Please clarify if these are trained with t0+210 or with mixed time data?

They are trained with $t_0 + 210$ minutes data. This has been clarified in the manuscript.

Figures 8 and 9, main text: The ordering, going from the top row of Fig8 to two rows of Fig9, back to the bottom row of Fig8, and then to the last row of Fig9 is counter-intuitive. This could surely be improved.

The labelling of the feature sets has been changed (see the updated Table 1) and the description of classification results with different features has been improved.

What F7 and F8 (old feature set labelling, F4 and F5 with the new labelling) add to the the feature list is mentioned in the paragraph before L367 in the old manuscript. If the comment refers instead to the rationale for adding F7 and F8 to the feature list, the idea is to verify if quantities which are meaningful for the human observer, such as Mach number, Alfven speed, plasma beta, hinder or help the unsupervised classification. As per the tests presented, the difference is not major.

L416: More comparison with Amaya(2020) of the results and potential future avenues would be good - it was only very briefly cited before.

This has been added to the conclusions:

*In Amaya 2020, more advanced pre-processing techniques were experimented with, which will most probably prove useful when we will move to the more challenging environment of spacecraft observations (as opposed to simulations). Furthermore, Amaya 2020 employed windows of time in the classification, which we have not used in this work in favor of an "instantaneous" approach. In future work, we intend to verify which approach gives better results.*

L447: Do you have any references to indicate what direction non-linear feature correlation analysis in deciding a dimensionality reduction could take? What about the addition of non-local features, such as spatial and temporal derivatives, curls etc?

We will try non-linear feature correlation analysis as part of future work. At the moment, we are not in the position to speculate on the results of this activity
Regarding the addition of non local features: in this work, we preferred a local, instantaneous approach, which gave good results with identification of large-scale magnetospheric regions. It may be unavoidable to introduce spatial and temporal derivatives if we decide to use a similar approach for the identification of boundaries between magnetospheric regions, or meso- and small-scale processes.

L448: Are dynamic SOMs a reasonable approach to automatic classification, or do they require user validation after every re-learning?

If the learning parameters are the same, the net casted by the DSOM has the same rigidity for any random initialization of the initial code word (node) locations. The final maps can potentially show slightly different arrangements, but as a net the SOM will cover the same area of the N-dimensional space. This has not been tested thoroughly in this paper but will be added to our future analysis.

**Referee 2**

In the manuscript under review the authors have applied unsupervised machine learning algorithms to analyse global magnetospheric simulation data obtained from OpenGGCM-CTIM-RCM code. The automated classification process uses principal component analysis for input data dimension reduction, self-organizing maps for training of an artificial neural network and K-means for cluster

extraction from the trained neural network. These unsupevised machine learning algorithms offer an automated way to determine clusters in the physical simulation data. The results shown in the paper are surely of great interest to space physics. The paper also displays the advantages and performance of the unsupervised clustering algorithm.

The paper contains interesing results, but contains no new method development, rather it presents an automated algorithm comprised of multiple unsupervised clustering algorithms. The application of methods is in overall well executed and explained.

There are points for revision that need to be addressed before advising for publication and minor suggestions for consideration. They are listed as following.

Major comments:

1. The authors do not discuss the stochastic nature of artificial neural networks, their sensitvity to initial conditions and convergence to local minima. Clarification for this issue would be needed to the section discussing Self Orgainzing Maps.

To clarify the issue, the following sentences have been to the manuscript:

*It is useful to remark that, even if the same data are used to train different SOMs, the trained networks will differ due e.g. to the stochastic nature of artificial neural networks and to their sensitivity to initial conditions. If the initial positions of the map nodes are randomly set (as in our case), maps will evolve differently, even if the same data are used for the training.*
*To verify that our results do not correspond to local minima, we have trained different maps seeding the initial random node distribution with different seed values. We have verified that the trained SOMs so generated give comparable classification results, even if the nodes that map to the same magnetospheric points are located at different coordinates in the map. The reason for this comparable classification results is that the 'net' created by a well-converged SOM will always have a similar coverage and neighbouring nodes will always be located at similar distances with respect to their neighbours (if the training data do not change). Hence, while the final map might look different, the classes and their properties will produce very similar end results. We refer the reader to Amaya et al. (2020) for exploration of the sensitivity of the SOM method to the parameters and to initial condition, and for a study of the rate and speed of convergence of the SOM.*

2. The nature of the hyper-parameters of the Self Organizing Maps are only briefly discussed in Section 3, more clarification on their influence to the algorithms performance is needed. A case study is done in appendix A, but a more theoretical description of the nature of the hyper-parameters should be added.

We have expanded Section 3 to better describe the ideas behind SOMs and the nature of the hyper-parameters.

*SOMs aim at producing an ordered representation of data, which in most cases has lower dimensionality with respect to the data itself. "Ordered" is a key word in SOMs. The topographical*

*relations between the trained SOM nodes are expected to be similar to those of the source data: data points that map to "nearby" SOM nodes are expected to have "similar" features. Each SOM node then represents a local average of the input data distribution, and nodes are topographically ordered according to their similarity relations [Kohonen 2014].*

*...*

*It is useful to compare the learning procedure in SOMs and in another, perhaps more known, unsupervised classification method, K-means [Lloyd 1982]. Both SOMs and K-means classification identify and modify the best matching unit for each new input. In K-means, only the winner node is updated. In SOMs, the winner node and its neighbors are updated. This is done to obtain an ordered distribution: nearby nodes, notwithstanding their initial weights, are modified during training as to become more and more similar.*

*...*

*The radius of the neighboring function $\sigma(\tau)$ determines how far from the winning node the update introduced by the new input will extend. The learning rate $\epsilon(\tau)$ gives a measure of the magnitude of the correction. Both are slowly decreased with the iteration number. At the beginning of the training, the update introduced by a new data input will extend to a large number of nodes (large $\sigma$), which are significantly modified (large $\epsilon$), since it is assumed that the map node do not represent well the input data distribution. At large iteration number, the nodes are assumed to have already become more similar to the input data distribution, and lower $\sigma$ and $\epsilon$ are used for "fine tuning". In this work, we choose to decrease $\sigma$ and $\epsilon$ with the iteration number. Another option, which we do not explore, is to divide the training into two stages, coarse ordering and final convergence, with different values of $\sigma$ and $\epsilon$.*
*However small, $\sigma$ has to be kept larger than 0, otherwise only the winning node is updated, and the SOM loses its ordering properties [Kohonen 2014].*

3. In Section 3 the authors do not mention what distance metric is used for the matching rule in Equation (1) and updating rule in Equation (2) of the SOM.

Euclidean norm, added to the manuscript.

4. Clarification should be added about how the authors initialized the neural map of the SOM.

The initial nodes are randomly distributed. We have verified that SOMs trained with the same data starting from different initial node distribution give comparable results.

This sentence has been added to the manuscript:

*Our maps are initialized with random node distributions. It has been demonstrated that different initialization strategies, such as using as initial node values a regular sampling of the hyperplane spanned by the two principal components of the data distribution, significantly speed up learning [Kohonen 2014].*

5. The influence of sampling of input data during learning needs to be discussed.

This issue has been clarified in the manuscript as follows:

*The selection of these points is randomized, and the seed of the random number generator is fixed to ensure that results can be reproduced. Tests with different seeds and with an higher number of training points did not give significantly different classification results.*

6. The authors should describe how much confidence they have in the result of the SOM.

As we have described in the revised Conclusions, we are at the moment quite happy with the classification results we obtain, because they map well to our knowledge of the system and appear to be quite robust to temporal variations in the simulated magnetosphere. The fact that a good subset of the features we tested gives comparably good results also points in the direction of a robust procedure on simulated data. Of course the real test for the method will be using spacecraft data, which will be extremely more challenging in terms of instrument noise, instrument limitations, presence of kinetic processes.

7. In Section 4.2 the model validation is done only by visual inspection. The colors of similar clusters differ from image to image. This fact also makes the readability of the figure very hard, as the eye automatically matches colors. A quantitative measure should be introduced for the robustness of the SOM. To discuss Figure 7 data, one could simply calculate for each result comparison pair for the same data a percentage of similarly labelled data points. Human labelling of the 7 clusters is already previously done (Figure 4 panel (d)). The same labeling system could be used for the K-means classification results displayed on Figure 7.

The colors in all plots have been matched to those of F1, to simplify visual comparisons of results.

We have added quantitative estimate of the agreement of the SOMS+ K-means vs pure K-means classification in Section 4.2.2:

*To compare the two classification methods quantitatively, we calculate the number of points which are classified in the same cluster with SOMs plus K-means vs pure K-means classification. 92.15 % of the points are classified in the same cluster, 92.74 % if the two magnetosheath clusters just downstream the bow shock are considered the same. These percentage are calculated on the entire training dataset at time $t_0 + 210$ minutes, of which cuts are depicted in the panels in Figure 7.*

8. Clarify what time snapshots and data is analysed on Figure 8. Do the panels on the figure correspond to the same dataset with different training features? Clusters depicted on Figure 8 follow different coloring schemes, which diminish the readability of the results. To better quantify the comparison of independent algorithm results on the same data a quantitative measure should be added. Similarly to the previous comment 7, Figure 4 panel (d) labeling system can be applied to all results. One could count the fraction of similarly labelled input data points for each compared pair of independent results for the same data.

In the submitted manuscript, Figure 8 depicted classification with different sets of training features, depicting the classification of training data points. In the new version, we decided to depict results for the same validation dataset, as Figure 6. In both cases, $t = t_0 + 210$ minutes. The colors have been changed to match the cluster description in Figure 4.

We have added Table 3, where we show, for each of the feature set examined, how many of the points in the validation dataset at time $t_0 + 210$ minutes are classified as in T1 (column S). We also examine the case when the two magnetosheath clusters downstream the bow shock are merger (column M).

The manuscript has been edited as follows:

*In Table 3, second column ("S"), we report the percentage of data points classified in the same cluster as F1 for each of the feature sets of Table 1, for the validation dataset at $t_0 + 210$ minutes. In the third column ("M"), we consider cluster 1 and 4 as a single cluster: in the previous analysis, we remarked that cluster 1 and 4 (the two magnetosheath clusters just downstream the bow shock) map to the same kind of plasma. We keep this into account when comparing classification results with F1.*

*The metrics depicted in Table 3 cannot be used to assess the quality of the classification per se, since we are not comparing against ground truth, but merely against another classification experiments. However, it gives us a quantitative measure of how much different classification experiments agree.*

9. Similar quantitative robustness measure should be added for the comparison of independent results for the same data in Figure 9.

See table 3.

10. Use the same coloring scheme for all cluster analysis results (Fig 6 – Fig 9) as different colors lose information in Figures. In Figure 4 a labelling system with colors is created, which could be used for all figures. Only visual comparison of clusters over multiple images throughout the paper with different color coding is tiresome.

Done.

11. The comparison of panels between Figure 4, Figure 6, Figure 7, Figure 8 and Figure 9 would be easier if all the captions of the panel names would contain the time snapshot information. Panels on the Figure 6 have different sizes, panel sizes on a Figure should be uniform.

The time snapshot information have been added to all captions, and labels have been made uniform. We have also add (T) and (V) to the caption, to label the training and validation datasets.

Minor suggestions and questions:

L167: Does the dimension of the input data influence the training time of the map considerably? Are there other metrics that could possibly influence it more? Can the PCA be used to intialize the neural map of the SOM? Is the main motivation for input data dimensionality reduction to have more reliable results from training of the SOM?

In our experience, it is the number of points, rather than the number of features, that influences the training time the most, so the main motivation for dimensionality reduction was in fact not training time (even if that helped) but rather, as the reviewer suggests, generating a more reliable training dataset. The correlation analysis (and previous knowledge) made clear that several of the magnetospheric variables in fact carry the same information on the state of the plasma, and we aimed at compressing that information in a lower number of features, while preserving a high percentage of the variance of the original dataset.

PCA can certainly be used to initialize the SOM map, but in this case we chose random initialization. We will check if PCA initialization reduces training time significantly, as expected.

Fig 3: the name of panel (a) is in different size.
Fig 4: the style of panel descriptions differ in the Caption. A more uniform style would increase the readability of the figure.
Fig 7: the name of panel (b) is in different size.

We have improved figure presentation in the revised version.

---

## Author Response (AR2)

**Response to the reviewers**

We thank the reviewers for this second feedback on our work. We reply here to the comments of Referee 1.

**Referee 1**

The authors have commendably amended and improved the manuscript. I am happy to recommend publication with only a few minor corrections requested.

L101: Please confirm the -X extent of the simulation. Do the cells stretch that much in the X-direction, or is the extent in fact -300 RE or similar?

We confirm that the simulation domain is indeed -3000 $R_E$ in the anti-Sunward direction.

L237 and Table 2: should not the middle set be F8 (B clipped) in accordance with the re-naming of feature sets? (also later at e.g. L437)

Indeed. We have edited the manuscript accordingly.

L240: Please at least briefly mention the use of F9 in the table - for that feature set, both PC1 and especially PC2 are impacted my magnetic field magnitude. It is currently down on line 468 so stating that you'll discuss the latter part of the table towards the end of section 4.3 should be enough - or you could move that bit here, especially as it actually discusses the violin plots, not the validation results.

The discussion has been moved to L240 as suggested.